# De novo MYC addiction as an adaptive response of cancer cells to CDK4/6 inhibition

Míriam Tarrado-Castellarnau[1,2], Pedro de Atauri[1,2,†] [ID], Josep Tarragó-Celada[1,2,†], Jordi Perarnau[1,2], Mariia Yuneva[3], Timothy M Thomson[4] & Marta Cascante[1,2,*] [ID]

## Abstract

Cyclin-dependent kinases (CDK) are rational cancer therapeutic targets fraught with the development of acquired resistance by tumor cells. Through metabolic and transcriptomic analyses, we show that the inhibition of CDK4/6 leads to a metabolic reprogramming associated with gene networks orchestrated by the MYC transcription factor. Upon inhibition of CDK4/6, an accumulation of MYC protein ensues which explains an increased glutamine metabolism, activation of the mTOR pathway and blunting of HIF-1α-mediated responses to hypoxia. These MYC-driven adaptations to CDK4/6 inhibition render cancer cells highly sensitive to inhibitors of MYC, glutaminase or mTOR and to hypoxia, demonstrating that metabolic adaptations to antiproliferative drugs unveil new vulnerabilities that can be exploited to overcome acquired drug tolerance and resistance by cancer cells.

**Keywords** [13]C metabolic flux analysis; CDK4/6; glutaminase; MYC; tumor metabolic reprogramming

**Subject Categories** Cancer; Metabolism; Signal Transduction

**Mol Syst Biol. (2017) 13: 940**

## Introduction

Cell cycle regulation is frequently altered in cancer (Malumbres & Barbacid, 2009). In particular, cyclin-dependent kinases 4 and 6 (CDK4/6) have been studied as therapeutic targets in cancer due to their essential role in the regulation of cell cycle progression at the G1 restriction point (Malumbres & Barbacid, 2009; Peyressatre *et al*, 2015). Highly selective and potent small-molecule CDK4/6 inhibitors have been developed and approved for clinical development (Fry *et al*, 2004; Finn *et al*, 2009; Leonard *et al*, 2012; Beaver *et al*, 2015). These new generation inhibitors effectively impair the growth of tumors with a strong reliance on CDK4/6-Cyclin D1 (Peyressatre *et al*, 2015; O'Leary *et al*, 2016). Among these inhibitors, palbociclib (PD0332991)

has shown the most promising results in phase III studies (Beaver *et al*, 2015; O'Leary *et al*, 2016). However, mechanisms of acquired resistance to CDK4/6 inhibitors are beginning to emerge which include loss of *RB1* or *CCNE1* amplification (Herrera-Abreu *et al*, 2016).

In spite of the large body of knowledge that has been amassed about cell cycle regulation and the increasing recognition of the importance of metabolic changes associated with cell cycle progression (Moncada *et al*, 2012; Diaz-Moralli *et al*, 2013), relatively little is known of the metabolic adaptations undergone by cancer cells in response to pharmacological inhibition of cell cycle regulators. To extend the benefits of CDK4/6 inhibition and to mitigate acquired resistance in cancer management, the adaptations of cancer cells to CDK4/6 inhibition need to be investigated by exploring not only altered signaling pathways but also their crosstalk with metabolic reprogramming. In this regard, tumor metabolic reprogramming is emerging as a key component of adaptive responses to drug-induced stress that may unveil novel actionable vulnerabilities of cancer cells (Maiso *et al*, 2015; Tarrado-Castellarnau *et al*, 2016). Specifically, recent work by Franco *et al* has unveiled metabolic reprogramming events and actionable metabolic targets, in particular mTOR, in pancreatic cancer cells in response to palbociclib (Franco *et al*, 2016).

Herein, we have undertaken a systematic study of the consequences on central carbon metabolism of the depletion or inhibition of CDK4/6 in cancer cells. By complementing metabolic analysis with transcriptomic data, we accurately depict relevant metabolic shifts associated with CDK4/6 depletion, revealing that the upregulation of MYC and its downstream network, which includes glutaminolysis and mTOR signaling, is both a direct consequence of, and a key adaptation to CDK4/6 inhibition.

## Results

### CDK4/6 depletion enhances glucose, glutamine, and amino acid metabolism

With the aim of elucidating metabolic adaptive responses and the emergence of potential vulnerabilities consequent to CDK4/6 depletion or inhibition, we targeted CDK4/6 in HCT116 human colorectal

---

1   Department of Biochemistry and Molecular Biomedicine, Faculty of Biology, Universitat de Barcelona, Barcelona, Spain
2   Institute of Biomedicine of Universitat de Barcelona (IBUB) and CSIC-Associated Unit, Barcelona, Spain
3   The Francis Crick Institute, London, UK
4   Institute of Molecular Biology of Barcelona, National Research Council (IBMB-CSIC), Barcelona, Spain
    *Corresponding author. Tel: +34 93 4021593; E-mail: martacascante@ub.edu
    †These authors contributed equally to this work

carcinoma cells by using specific small interference (si) RNAs. These cells bear a loss-of-function p16[INK4a] mutant allele and a wild-type allele silenced through a hypermethylated promoter, resulting in full loss of functional p16[INK4a] (Myohanen *et al*, 1998), which can lead to a higher activation status of CDK4/6. Double CDK4/6 knockdown in HCT116 cells (Fig 1A) was more effective in reducing cell proliferation than depleting CDK4 or CDK6 alone (Fig 1B), confirming that these kinases play compensatory roles (Malumbres *et al*, 2004). CDK4/6 inhibition caused an increase in cell volume but had no effect on the total cellular protein content (Fig EV1A–D). In association with a reduced proliferation, knockdown of CDK4/6 promoted early apoptosis (Fig 1C) and an expected accumulation of cells in the G1 phase of the cell cycle (Fig 1D). Equivalent results were obtained in HCT116 cells treated with the selective CDK4/6 inhibitor PD0332991 (Fig EV1E–L).

Unexpected for the observed cell proliferation inhibitory activity, knockdown of CDK4/6 or inhibition with PD0332991 enhanced glucose and glutamine consumption as well as lactate and glutamate production (Fig 1E), accompanied with greater production rates of aspartate, asparagine, proline, glycine, and alanine and higher consumption rates of serine, arginine, cysteine, threonine, isoleucine, leucine, and lysine (Fig 1F).

Cells depleted of CDK4/6 displayed higher extracellular acidification rate (ECAR) than control cells (Fig 1G), a reflection of higher production of lactate derived from glycolysis. Real-time ECAR determinations while adding glucose and oligomycin to cells cultured in glucose-deprived medium identified an increase in glycolysis, glycolytic capacity, and glycolytic reserve associated with CDK4/6 knockdown (Fig 1H). Further evidence indicated that CDK4/6 depletion favors glycolytic flux through the non-oxidative branch of the pentose phosphate pathway (PPP) (Appendix Fig S1). In spite of their higher glycolytic activity, CDK4/6-kd cells were less sensitive to changes in glucose availability than control cells (Fig 1I), suggesting that they may overcome glucose deprivation by increasing alternative metabolic pathways such as glutaminolysis or glycogenolysis (Pelletier *et al*, 2012). Indeed, CDK4/6-kd cells presented higher glycogen storage than control cells (Fig 1J).

### CDK4/6 depletion enhances mitochondrial metabolism and function

We next examined through [13]C-isotope labeling experiments the contribution of glucose to the tricarboxylic acid (TCA) cycle by comparing the ratios of m2-labeled metabolites to m2 pyruvate, as m2 is the most abundant isotopologue (mass isotopomer) derived from [1,2-[13]C$_2$]-glucose (Grassian *et al*, 2011) (Fig 1K, left). CDK4/6-kd cells presented higher ratios of m2 citrate, glutamate, α-ketoglutarate, malate, and aspartate, indicating a greater incorporation of labeled pyruvate into the TCA cycle than control cells (Fig 1K, right).

Because CDK4/6-kd cells consumed more glutamine and produced more glutamate than control cells (Fig 1E), we examined the fate of glutamine carbons by monitoring uniformly labeled

---

**Figure 1. Effects of CDK4/6 knockdown on growth and glucose and glutamine metabolism of HCT116 cells.**

A   Efficacy of siRNA-mediated knockdown of CDK4 and CDK6 in HCT116 cells, assessed by Western blotting.

B   Effect of CDK4 and/or CDK6 knockdown on cell proliferation.

C   Early apoptosis percentage in CDK4/6-kd and control cells assessed by flow cytometry.

D   Cell cycle distribution in CDK4/6-kd and control cells determined by flow cytometry.

E   Comparative extracellular metabolic fluxes for CDK4/6-kd and control cells. Glucose and glutamine consumption and lactate and glutamate production rates were obtained after 24 h of incubation with fresh media and normalized to cell number.

F   Non-essential (left) and essential (right) amino acid consumption and production rates in CDK4/6 knockdown and control cells obtained after 24 h of incubation with fresh media and normalized to cell number. Asp, aspartate; Ser, serine; Asn, asparagine; Pro, proline; Gly, glycine; Ala, alanine; Arg, arginine; Cys, cysteine; Thr, threonine; Ile, isoleucine; Leu, leucine; Lys, lysine.

G   Basal extracellular acidification rates (ECAR) normalized to cell number.

H   ECAR profiles illustrating glycolysis, glycolytic capacity, and glycolytic reserve in control (left) and CDK4/6 knockdown cells (right). Cells were incubated in the absence of glucose, and sequential injections of glucose and oligomycin were performed. Data are normalized to cell number and represented as mean ± SD from $n = 5$.

I   Glucose dependence of CDK4/6-kd and control cells. Cultures were deprived of glucose (−Glc) for 72 h, and the effect on cell proliferation was determined.

J   Glycogen quantification in CDK4/6-kd and control cells normalized to cell number.

K   Glucose contribution to the TCA cycle in CDK4/6-kd and control cells. Cells were incubated in the presence of 10 mM [1,2-[13]C$_2$]-glucose for 24 h, and TCA intermediates were isolated from cells for isotopologue distribution analysis. Left, schematic representation of the labeling distribution in TCA intermediates from [1,2-[13]C$_2$]-glucose in the first turn of the TCA cycle. The incorporation of [13]C-labels to the TCA intermediates is considered to be via pyruvate dehydrogenase (PDH), pyruvate carboxylase (PC), and malic enzyme (ME2). For clarity purposes, only the first turn of the TCA cycle is represented. Right, m2 citrate, m2 glutamate, m2 α-ketoglutarate (αKG), m2 malate, and m2 aspartate labeling ratios normalized to m2 pyruvate (pyr) labeling.

L   Glutamine contribution to the TCA cycle in CDK4/6-kd and control cells. Cells were incubated in the presence of 2 mM [U-[13]C$_5$]-glutamine for 24 h, and TCA intermediates were isolated from cells for isotopologue distribution analysis. Schematic representation of the labeling distribution in TCA intermediates from [U-[13]C$_5$]-glutamine in the first turn of the TCA cycle considering the oxidative TCA cycle (red labeling) and the reductive carboxylation (green labeling). The incorporation of [13]C-labels to the TCA intermediates is considered to be via glutaminase and glutamate dehydrogenase. For clarity purposes, only the first turn of the TCA cycle is depicted.

M   Glutamine contribution to the TCA cycle in CDK4/6-kd and control cells. Right, normalized m4 citrate, m5 glutamate, m5 α-ketoglutarate (αKG), m4 malate, and m4 aspartate labeling are indicative of the oxidative TCA pathway. Values are normalized to total label enrichment (Σm). Left, normalized m5 citrate, m3 aspartate, and m3 malate are obtained through the reductive carboxylation of [U-[13]C$_5$]-glutamine. Values are normalized to total label enrichment (Σm).

N   Quantification of intracellular metabolites related to mitochondrial metabolism (mean ± SD of $n = 4$). Data are normalized by cell number and represented as a percentage relative to control cells for each metabolite.

Data information: CDK4/6, CDK4/6-kd cells; Control, cells transfected with non-targeting RNA duplexes. All experiments were performed 96 h after siRNA transfection or PD0332991 treatment. Shown values are mean ± SD for $n = 3$ (except where otherwise indicated). Significance was determined by two-tailed independent sample Student's *t*-tests. Statistically significant differences between CDK4/6-inhibited and control cells are indicated as $P < 0.05$ (*), $P < 0.01$ (**), and $P < 0.001$ (***), while differences between treatment (glucose deprivation) and the corresponding control are shown as $P < 0.05$ (#) for CDK4/6-inhibited cells and as $P < 0.01$ (¶¶) for control cells.

Source data are available online for this figure.

[U-$^{13}$C$_5$]-glutamine incorporation into TCA cycle intermediates (Fig 1L). Direct glutamine contribution to citrate (m4), glutamate (m5), α-ketoglutarate (αKG) (m5), malate (m4), and aspartate (m4) was significantly increased in CDK4/6-kd cells, demonstrating an enhanced glutamine oxidation and consequent carbon contribution to the oxidative TCA cycle (Fig 1M, right). CDK4/6-kd cells presented higher levels of m5 citrate, m3 aspartate, and m3 malate (Fig 1M, left), revealing a more active reductive carboxylation of m5 α-ketoglutarate (Metallo *et al*, 2012; Fendt *et al*, 2013). Nevertheless, the relative proportion of m5 citrate, m3 aspartate and m3 malate versus m4 citrate, m4 aspartate, and m4 malate-labeled species indicated that oxidative TCA cycle was the main pathway of glutamine metabolism in both cases. Further, the concentration of

TCA intermediates and related metabolites was significantly increased in CDK4/6-kd cells (Fig 1N).

CDK4/6-kd cells exhibited higher oxygen consumption rates (OCR) than control cells (Fig 2A and B), indicating an augmented mitochondrial respiration which is in agreement with recently published studies (Franco *et al*, 2016). Under glutamine deprivation, CDK4/6-kd cells maintained near-control levels of basal respiration in complete medium (Fig 2C) and decreased the levels of acidification (Fig 2D), evidencing a shift of glucose metabolism into the TCA cycle and away from lactate production in the absence of glutamine. Under glucose deprivation, OCR was increased by 2-fold (Fig 2C), while a drop in ECAR was observed in CDK4/6-kd cells (Fig 2D), indicating that enhanced glutamine mitochondrial

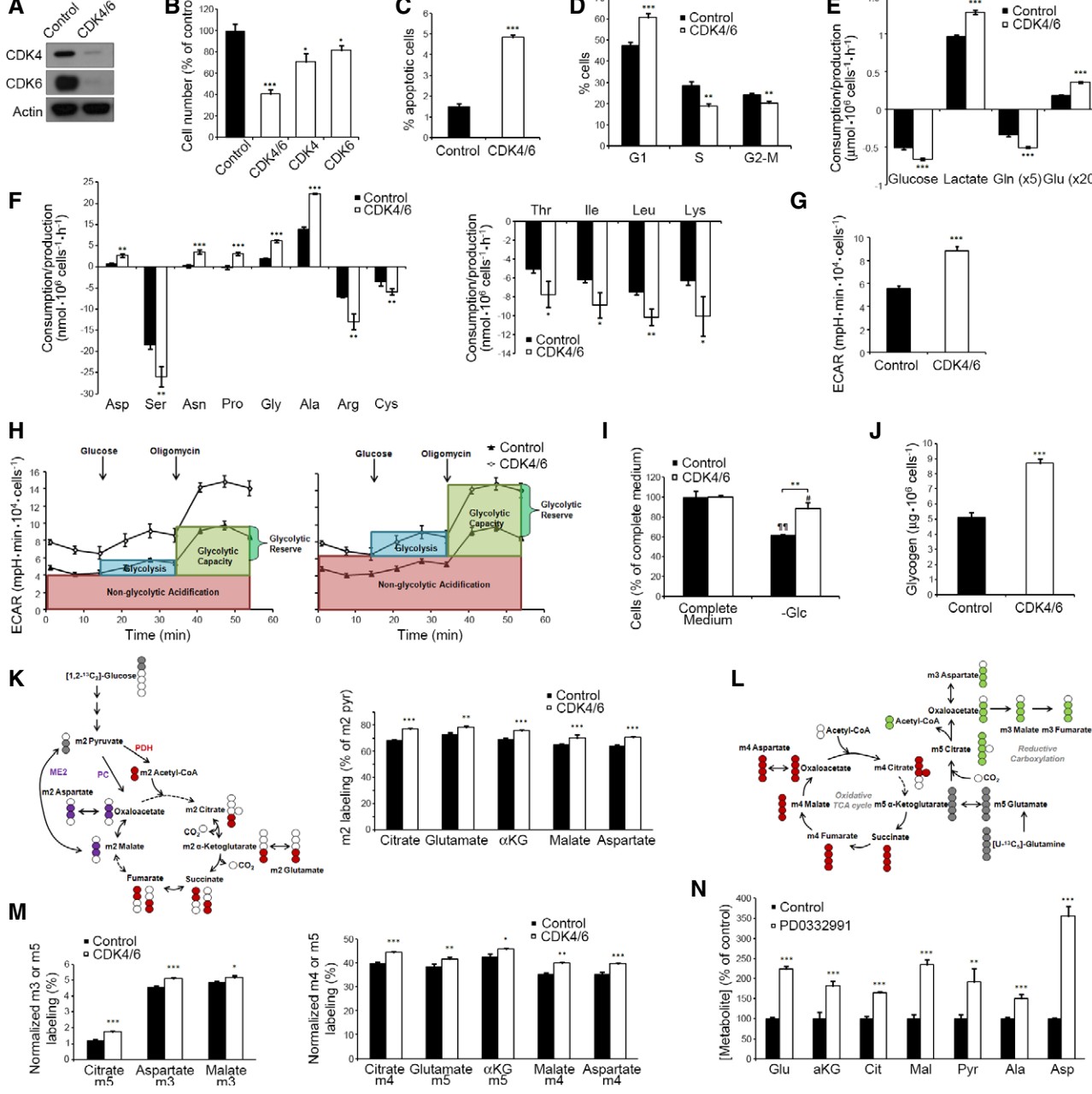

**Figure 1.**

metabolism may compensate for the reduced glycolytic ATP production. Additional experiments evidenced that CDK4/6-kd cells shifted to glutamine as the preferred substrate for respiration (Fig 2E–H), thereby becoming more sensitive than control cells to the electron transport chain component inhibitors FCCP (Fig 2I, left) and rotenone + antimycin A (Fig 2I, right).

Collectively, these observations indicate that CDK4/6 knockdown increases mitochondrial metabolism through elevated utilization of glutamine and enhanced mitochondrial respiratory capacity and are in agreement with Franco et al (2016) results for a pancreatic cancer cell model. As such, specific metabolic reprogramming events in response to CDK4/6 depletion or inhibition appear to be conserved among cancer cells of different origin. Additional experiments showed that CDK4/6 depletion increased glutathione, NADPH, and ROS levels,

while it impaired fatty acid synthesis in HCT116 cells (Fig EV2), all of which are processes where glutamine is or can be involved.

### A quantitative model of central carbon metabolism in CDK4/6-depleted HCT116 cells

To infer differential intracellular metabolic flux distributions in CDK4/6-kd versus control cells, we constructed a quantitative metabolic network model of central carbon metabolism and performed a $^{13}C$ metabolic flux analysis (Niedenfuhr et al, 2015) by applying the MATLAB-based software package INCA (Young, 2014). For this, we combined direct extracellular measurements, such as oxygen consumption, metabolite consumption and production rates, and protein synthesis rate, with isotopologue distributions in several

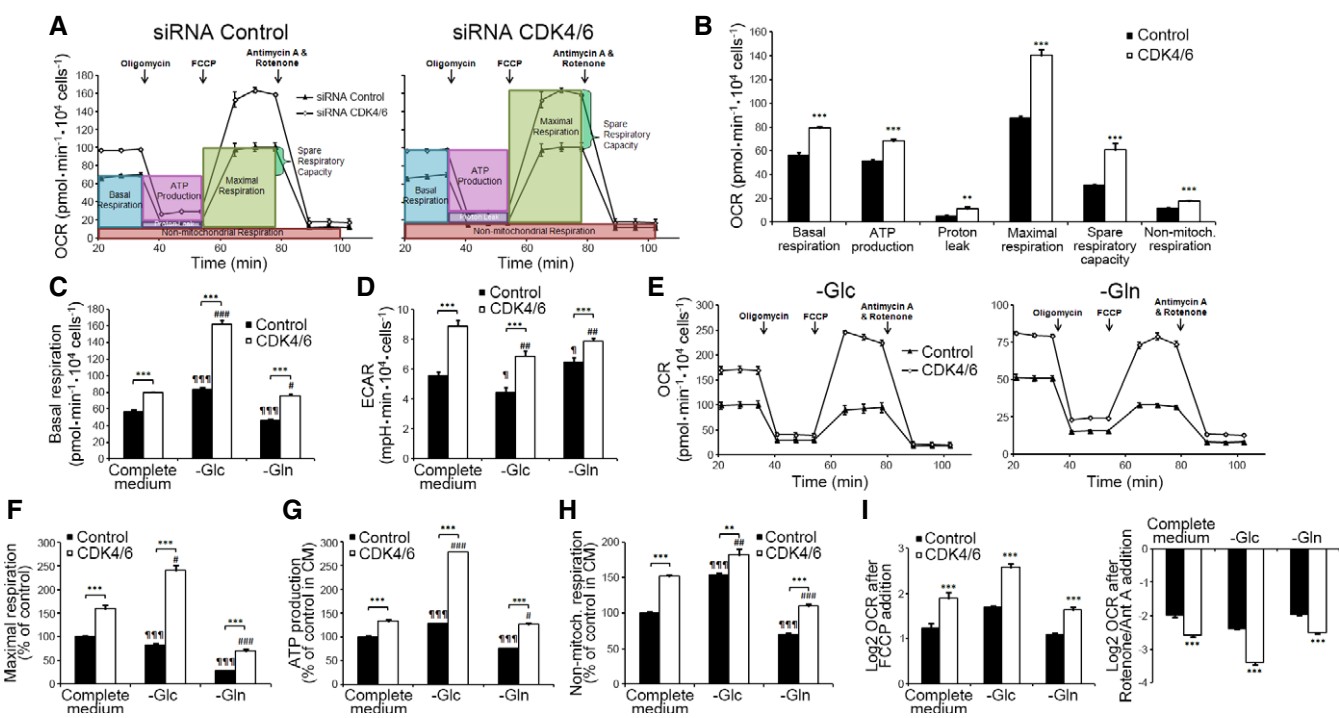

**Figure 2. Effect of CDK4/6 knockdown on the mitochondrial function of HCT116 cells.**

A   Oxygen consumption rate (OCR), following sequential injections of oligomycin (1.5 μM), FCCP (1 μM), and antimycin A (1 μM) + rotenone (1 μM), in control and CDK4/6-kd cells.

B   Quantification of OCR for basal respiration, ATP production-associated respiration, non-ATP linked oxygen consumption (proton leak), maximal respiration, spare respiratory capacity, and non-mitochondrial respiration, in complete medium and normalized to cell number.

C   OCR basal levels in complete and restricted (glucose- or glutamine-deprived) media conditions, normalized to cell number.

D   ECAR basal levels under the conditions summarized in (C).

E   Effects of glucose (left) and glutamine (right) deprivation on the mitochondrial function of CDK4/6-kd and control cells. OCR profiles were acquired following injections of oligomycin (1.5 μM), FCCP (1 μM), antimycin A (1 μM) + rotenone (1 μM).

F   Glucose and glutamine deprivation effects on maximal respiration.

G   ATP production-associated respiration.

H   Non-mitochondrial respiration.

I   Effects of mitochondrial uncoupling and mitochondrial complex I and III inhibition on OCR after FCCP addition (left) or rotenone and antimycin A (right) addition in complete and restricted media, expressed as $Log_2$ (fold change). Results are normalized to cell number and shown as percentage of OCR relative to non-targeting siRNA-treated cells cultured in complete medium. Cells were cultured for 1 h in glucose- or glutamine-deprived media (–Glc and –Gln, respectively) before experimental determinations were made, and these conditions were maintained throughout the experiment.

Data information: CDK4/6, CDK4/6-kd cells; Control, cells transfected with non-targeting RNA duplexes. Bars correspond to mean ± SD (n = 4). Significance was determined by two-tailed independent sample Student's t-tests. Statistically significant differences between CDK4/6 knockdown and control cells are indicated as $P < 0.05$ (*), $P < 0.01$ (**), and $P < 0.001$ (***), while differences between treatment (oligomycin, glucose, or glutamine deprivation) and the corresponding control (CDK4/6 siRNA- or non-targeting siRNA-treated cells in complete medium) are shown as $P < 0.05$ (#), $P < 0.01$ (##), and $P < 0.001$ (###) for CDK4/6-inhibited cells and as $P < 0.05$ (¶), $P < 0.01$ (¶¶), and $P < 0.001$ (¶¶¶) for control cells.

metabolites resulting from $^{13}$C propagation from labeled glucose and glutamine. The outcome of this analysis was compatible with all prior observations, supporting that CDK4/6-kd cells exhibit a higher mitochondrial activity accompanied with a higher dependence on glutamine and lower dependence on glucose (Fig 3 and Table EV1). In addition, the increased glutamine uptake and transformation to glutamate by glutaminase (GLS1) can also promote the mitochondrial respiratory activity since glutamate is the substrate for mitochondrial glutamate dehydrogenase (GDH). On the other hand, the extra demand of amino acids for protein synthesis matched the measured uptakes of essential amino acids in control cells, while CDK4/6-inhibited cells exhibited an extra uptake of several amino acids above the required value for protein synthesis (Appendix Fig S2).

### MYC, mTOR, and HIF-1α are key players in the adaptive cellular responses to CDK4/6 depletion

Gene expression profiling identified 1308 genes differentially expressed in CDK4/6-kd versus control cells (718 upregulated, 592 downregulated; Fig 4A; see also Table EV2). Gene set enrichment analysis (GSEA) (Subramanian et al, 2005) yielded a significant enrichment in CDK4/6-kd cells of signatures associated with MYC (upregulation), mechanistic target of rapamycin (mTOR) (upregulation), or hypoxia inducible factor 1α (HIF-1α) (downregulation) (Fig 4B; see also Table EV3). The inferred upregulation of the mTOR pathway is in agreement with previous studies (Franco et al, 2016). The expression levels of selected genes, including subsets encoding enzymes of the central carbon metabolism (Duarte et al, 2007; Schellenberger et al, 2010), were quantified by qRT–PCR. Consistent with the observed increased respiratory activity and mitochondrial function, genes associated with MYC and involved in glutamine and polyamine metabolism or with a role in mitochondrial biogenesis and function were overexpressed in CDK4/6-kd cells compared to control cells (Fig 4C) (Dang, 2013).

The above results suggest that CDK4/6 knockdown activates MYC-dependent functions. Indeed, CDK4/6 knockdown or inhibition with PD0332991 (Fig 5A; see also Fig EV4A) caused a significant accumulation of MYC as compared to control cells under standard growth conditions. Consistent with the known ubiquitin–proteasome-dependent turnover of MYC (Farrell & Sears, 2014), its relatively low levels in control cells were upregulated by treatment with the proteasome inhibitor MG132 (Fig 5A). This potential mode of degradation in a putative CDK4/6-dependent turnover of MYC was further supported by the observation of higher levels of poly-ubiquitin-conjugated MYC species in control than in CDK4/6-kd cells (Fig 5B). It has been recently reported that MYC peptides are substrates of the kinase activity of CDK4/6 (Anders et al, 2011). In order to test this hypothesis, we performed in vitro kinase assays with CDK4-Cyclin D1 or CDK6-Cyclin D1 complexes and full-length recombinant human c-MYC protein (Abcam, ab169901) as a substrate. Indeed, we detected specific $^{33}$P signals in both kinase reactions, indicating that both CDK4-Cyclin D1 and CDK6-Cyclin D1 complexes directly phosphorylate MYC (Fig 5D). With the purpose of determining the precise phosphorylation sites, we performed kinase assays with unlabeled ATP and analyzed MYC tryptic peptides by mass spectrometry. The results showed that peptides KFELLPT(phosphor)PPLSPSR and KFELLPTPPLS(phosphor)PSRR were phosphorylated on threonine 7 (corresponding to c-MYC T58)

and serine 11 (corresponding to c-MYC S62), respectively (Fig EV3A). Moreover, CDK4/6-kd cells displayed diminished P-MYC (Thr58)/MYC and P-MYC (Ser62)/MYC ratios compared to control cells (Fig 5C), supporting that phosphorylation of MYC at Thr58 and Ser62 is mediated by CDK4/6 in live cells. Consistently, cells expressing the MYC T58A phospho-resistant mutant mimicked the metabolic phenotype induced by CDK4/6 inhibition, as shown by enhancing glucose and glutamine consumption as well as lactate and glutamate production (Fig EV3B). Collectively, these observations suggest that CDK4/6-dependent phosphorylation is associated with the polyubiquitination and subsequent proteasomal degradation of MYC, thus offering a plausible mechanism for the accumulation of MYC upon inhibition of CDK4/6.

In agreement with the transcriptomic analysis, MYC upregulation in response to CDK4/6 knockdown or inhibition was accompanied with increased protein levels of glutaminase (GLS1) (Fig 5E; see also Fig EV4A), positively regulated by MYC through transcriptional repression of microRNA-23a/b (Gao et al, 2009). The GLS1 gene encodes two alternatively spliced isoforms, kidney glutaminase or KGA and glutaminase C or GAC (Mates et al, 2013). By using isoform-specific antibodies, we confirmed the overexpression of both GLS1 isoforms in CDK4/6-kd HCT116 cells (Fig EV4B), consistent with the increased glutamine consumption rates observed in CDK4/6-kd cells (Fig 1E). Further support for MYC activation in response to CDK4/6 depletion was provided by the observation of a significant upregulation, determined by qRT–PCR, of known MYC-regulated or associated genes, including GLS1, SLC7A5, SLC7A6, MAX, and genes of polyamine synthetic pathways (Fig 5F; see also Fig EV4C).

Our analysis additionally suggested that CDK4/6 knockdown induced an upregulation of the mTOR pathway (Fig 4; see also Table EV3), an inference experimentally supported by the observation of increased levels of P-mTOR (Ser2448) in CDK4/6-kd cells under standard growth conditions (Fig 5E), in concordance with recent studies (Franco et al, 2016). Intracellular glutamine acts upstream of mTOR complex 1 (mTORC1) serving as an efflux substrate for SLC7A5/SLC3A2, a plasma membrane heterodimer, to regulate the uptake of extracellular leucine, and leading to mTORC1 activation (Nicklin et al, 2009). Indeed, CDK4/6-kd cells exhibited enhanced glutamine and leucine uptake (Fig 1E and F) and upregulation of SLC7A5 and SLC3A2 (Fig 5F), thus providing robust evidence for this mechanism of activation of the mTOR pathway upon CDK4/6 knockdown. Furthermore, mTORC1 promotes the use of glutamine carbons through TCA anaplerosis by activating glutamate dehydrogenase (GDH) (Csibi et al, 2013). Augmented levels of TCA intermediates (Fig 1N) and the overexpression of GDH (Fig EV4B) supported that an enhanced glutamine mitochondrial metabolism followed CDK4/6 depletion. Further, downregulation of tuberous sclerosis (TSC) 2 and TSC1, key mTOR-negative regulators (Sabatini, 2006), was observed in CDK4/6-kd cells (Fig 5G). It has been shown that MYC activates the mTOR signaling pathway through direct repression of TSC2 (Ravitz et al, 2007). Thus, multiple mechanisms may underlie the upregulation of the mTOR pathway induced by the accumulation of MYC that follows CDK4/6 inhibition.

In addition, CDK4/6-kd cells displayed increased levels of P-Akt (Ser473) (Fig 5G), a modification catalyzed by mTORC2 (Sarbassov et al, 2005; Laplante & Sabatini, 2012). This was accompanied with a downregulation of p27$^{Kip1}$ (Fig 5G) and upregulation of PIK3R3

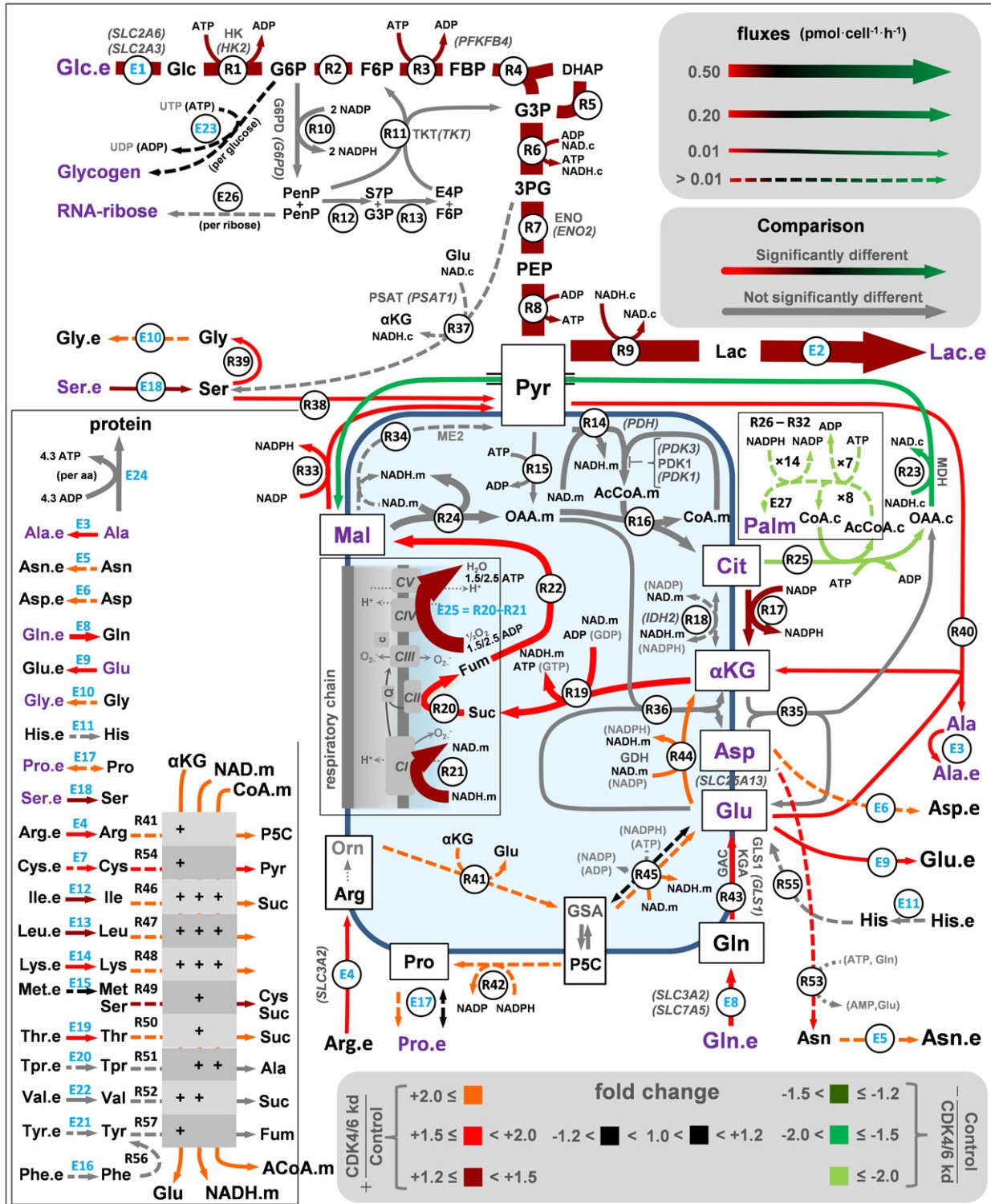

**Figure 3. Metabolic shifts following CDK4/6 knockdown in HCT116 cells.**

A quantitative $^{13}$C metabolic flux analysis was performed with experimental measures of label propagation from glucose and glutamine. The software package INCA was applied to estimate flux values for all these processes, which involved a fitting procedure of predicted and measured values for labeled products (in purple) and extracellular measured processes (in blue). Fold changes refer to CDK4/6-kd versus control cells and provide a measure of the changes in the flux calculated with the best fit values. Color in arrows (red, black, green) indicates fold changes for reactions whose 95% confidence intervals are not overlapping. Fold changes for the reaction fluxes whose 90% (R23 and R49) and 80% (E23, E27, R25-32, R38, and R45) confidence intervals are not overlapping are also included. For clarity purposes, some proteins (bold letters) and genes (italic letters in round brackets) are shown. See Appendix and Table EV1 for abbreviations and a full description of the model, data, and the resulting flux map.

(encoding the PI3K regulatory subunit p55γ) and *CCND1* (Cyclin D1) (Fig EV4D), known to be mediated through inactivation of FOXO3a by P-Akt (Biggs *et al*, 1999; Brunet *et al*, 1999).

The GSEA-predicted downregulation of hypoxia-associated responses upon CDK4/6 knockdown correlated with lower expression levels of HIF-1α and HIF-2α proteins in CDK4/6-kd cells than control cells in normoxia (Fig 5E; see also Fig EV4F). This was accompanied

with an upregulation of pyruvate dehydrogenase (PDH) (Fig 5H) and mitochondrial activities (Fig 5I), concomitant with a downregulation of PDH kinases PDK1 and PDK3 (Fig 5H), targets of the transcriptional activation of HIF-1α. The mitochondrial uptake of glutamate was also increased in CDK4/6-kd cells, concomitant with an upregulation of the glutamate transporter SLC25A13 (Fig EV4E). It has been shown that increased glutaminolysis generates higher levels of

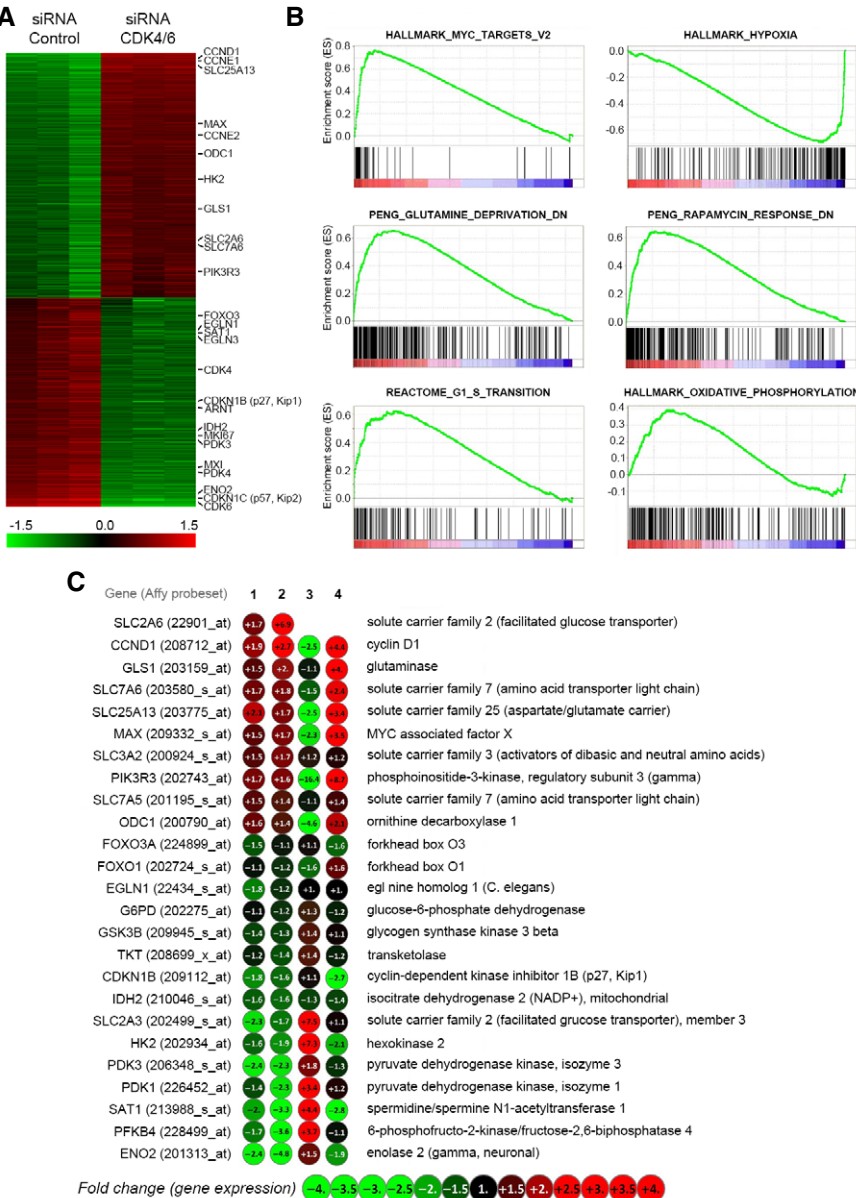

**Figure 4.  Transcriptomic analysis of CDK4/6 knockdown in HCT116 cells.**

A  Heatmap of differential gene expression of CDK4/6-kd versus control cells. Relative positions of selected genes are indicated on the right.

B  GSEA enrichment plots showing the correlation of the expression profiles of CDK4/6-kd versus control cells of gene sets upregulated by MYC, upregulated in response to hypoxia, downregulated after glutamine deprivation, downregulated in response to rapamycin, involved in G1/S transition, and involved in oxidative phosphorylation.

C  Expression levels of selected genes. Column 1, differentially expressed genes in CDK4/6-kd versus control identified from Affymetrix GeneChip analysis (Table EV2). Column 2, gene expression levels determined by qRT–PCR (CDK4/6-kd versus control). Column 3, effect of hypoxia (1% $O_2$) on gene expression in control HCT116 cells, determined by qRT–PCR (1% $O_2$ versus control). Column 4, effect of CDK4/6 knockdown on gene expression in HCT116 cells under hypoxia (1% $O_2$), determined by qRT–PCR (CDK4/6-kd + 1% $O_2$ versus control + 1% $O_2$). Data were blindly assessed.

intracellular α-ketoglutarate, which is a substrate of, and activates, prolyl hydroxylases (PHDs), triggering PHD-dependent HIF-1α hydroxylation and proteasome-mediated degradation, even under hypoxia (Fig 5E) (Tennant *et al*, 2009). Incubation of cells with the PHD inhibitor DMOG upregulated HIF-1α in CDK4/6-kd cells (Fig 5E), revealing that the observed reduction of HIF-1α levels in normoxia is mediated by PHD hydroxylation. Conversely, hypoxia caused a significant downregulation of MYC, GLS1 and P-mTOR (Figs 4C and 5E), partially countering the upregulation of MYC subsequent to CDK4/6 knockdown (Gordan *et al*, 2007).

Importantly, several of the observed responses to CDK4/6 knockdown or inhibition were not limited to HCT116 cells. Thus, treatment of MCF-7 or SK-BR-3 breast cancer cells with the CDK4/6 inhibitor PD0332991 also caused an upregulation of MYC, GLS1 and the mTOR and Akt signaling pathways, as well as a downregulation of HIF-1α (Fig 5J), suggesting that upregulation of MYC, with its consequent downstream effects, is an adaptive response to CDK4/6 loss of function conserved among several types of cancer cells.

Cells synchronized in the G1 phase of the cell cycle (Fig 5K) did not accumulate MYC, GLS1 or P-mTOR as compared to asynchronous cells, and presented higher HIF-1α levels than cells with CDK4/6 inhibition (Fig 5L). We found the same results with NCM460 human epithelial cells derived from healthy colon mucosa (Fig EV4G and H). Further, the consumption and production rates of glucose, glutamine, lactate and glutamate in G1-synchronized cells were significantly lower than those of control HCT116 cells (Fig 5M) and not significantly different from asynchronous NCM460 cells (Fig EV4I). These observations support the conclusion that the modulation of MYC, GLS1, P-mTOR, and HIF-1α that follows CDK4/6 depletion or inhibition is not a result of G1 arrest but directly attributable to a loss of function of CDK4/6.

## CDK4/6 depletion or inhibition sensitizes cells to the inhibition of MYC, glutaminase, mTOR or PI3K or to hypoxia

We reasoned that the upregulation of MYC and its downstream effector functions after CDK4/6 depletion or inhibition could make the cells addicted to MYC (Yoo *et al*, 2008). In agreement with previous studies (Lin *et al*, 2007), treatment with the MYC-MAX heterodimerization inhibitor 10058-F4 (Yin *et al*, 2003) strongly downregulated MYC protein levels causing a 60% decrease in control cell numbers and a complete abrogation of proliferation in CDK4/6-kd cells at 50 μM (Fig 6A). In addition, 10058-F4 treatment caused a dose-dependent inhibition of cell proliferation and a reversal of the effects of PD0332991 on GLS1, HIF-1α, P-mTOR, P-S6K, and P-Akt protein levels (Fig 6B). Likewise, specific knockdown of MYC with RNAi duplexes had the same effect as 10058-F4 treatment to synergize with CDK4/6 knockdown in reducing cell proliferation (Fig 6C) and to reverse the effects of CDK4/6 inhibition both on the consumption and production rates of glucose, glutamine, lactate and glutamate (Fig 6D) and on GLS1, HIF-1α, P-mTOR, P-S6K, and P-Akt protein levels (Fig 6E).

As described above, the major MYC target and metabolic effector glutaminase (GLS1) is significantly upregulated in CDK4/6-kd HCT116 cells. Treatment of cells with the specific GLS1 inhibitor bis-2-(5-phenylacetoamido-1,2,4-thiadiazol-2-yl)ethyl sulfide (BPTES) (Robinson *et al*, 2007) for 72 h selectively reduced the viability of CDK4/6-kd cells without significantly compromising that of control cells (Fig 6F). In contrast, glutamine deprivation

---

**Figure 5. CDK4/6 knockdown causes upregulation of MYC, GLS1, and P-mTOR and downregulation of HIF-1α.**

A   CDK4/6 knockdown induces an upregulation of MYC. Western blotting analysis of total protein fractions of control and CDK4/6-kd cells after incubation with the proteasome inhibitor MG132 or vehicle for 6 h.

B   CDK4/6 knockdown is accompanied with a lower abundance of polyubiquitinated MYC. Control and CDK4/6-kd cells were treated with or without the proteasome inhibitor MG132 for 6 h before collection for immunoprecipitation (IP). Samples were immunoprecipitated with MYC antibody and subjected to immunoblotting using an anti-ubiquitin antibody.

C   CDK4/6 knockdown is accompanied with decreased MYC phosphorylation. MYC, P-MYC Ser62, and P-MYC Thr58 protein levels were determined by Western blotting. Bands were quantified by densitometry analysis (bottom) using the ImageJ software and represented as mean band intensity of P-MYC/MYC ratio normalized to β-actin.

D   Kinase assays of CDK4/Cyclin D1 and CDK6/Cyclin D1 on full-length recombinant human MYC protein. Results are expressed as percentage of MYC phosphorylation as compared to RB phosphorylation by CDK4/Cyclin D1 and CDK6/Cyclin D1.

E   Effects of CDK4/6 knockdown on signaling pathways. Western blotting analysis of total protein fractions of CDK4/6-kd and control cells under normoxic or hypoxic (1% $O_2$) conditions or after DMOG treatment for 24 h.

F   Upregulation of GLS1, SLC7A6, SLC7A5, SLC3A2, and MAX in CDK4/6-kd cells. Gene expression was assessed by qRT–PCR. Results are normalized to cyclophilin A and expressed as fold change relative to control cells.

G   CDK4/6 knockdown induces activation of mTOR and Akt signaling pathways in HCT116 cells. Western blotting analysis of total protein fractions of CDK4/6-inhibited and control cells.

H   CDK4/6 knockdown induces activation of pyruvate dehydrogenase (PDH). Top, PDH activity in control and CDK4/6-kd cells. Bottom left, protein levels of PDH, P-PDH, and PDK1, determined by Western blotting. Bottom right, gene expression levels of PDK1 and PDK3, quantified by qRT–PCR.

I   Enhanced mitochondrial activity upon CDK4/6 knockdown in HCT116 cells estimated by MTT assays normalized by cell number.

J   Western blotting analysis of total protein fractions of CDK4/6-inhibited and control MCF-7 and SK-BR-3 breast cancer cells. Protein extracts were obtained after incubating MCF-7 and SK-BR-3 cells with PD0332991 (2 μM) or vehicle for 96 h.

K   Cell cycle distribution of HCT116 cells synchronized by serum deprivation.

L   Western blotting analysis of total protein fractions of CDK4/6-inhibited, serum-deprived, and control HCT116 cells.

M   Comparative extracellular metabolic fluxes for serum-starved and control cells. Glucose and glutamine consumption and lactate and glutamate production rates were obtained after 24 h of incubation with fresh media and normalized to cell number.

Data information: CDK4/6, CDK4/6-kd cells; Control, cells transfected with non-targeting RNA duplexes. Bars correspond to mean ± SD (*n* = 3). Statistically significant differences between CDK4/6-kd and control cells were determined by two-tailed independent sample Student's *t*-tests and are indicated as $P < 0.05$ (*), $P < 0.01$ (**), and $P < 0.001$ (***).

Source data are available online for this figure.

    

caused a significant reduction in the viability of both control and CDK4/6-kd cells. Importantly, the addition of the cell-permeable form of α-ketoglutarate (αKG), dimethyl α-ketoglutarate (dm-αKG), rescued the viability of BPTES-treated CDK4/6-kd cells while it had no effect on control cells (Fig 6F). These observations indicate that the glutamine and glutaminase dependence observed in CDK4/6-kd cells is not explained by the requirement of amide groups and nitrogen for the biosynthesis of nucleotides and non-essential amino acids, as dm-αKG is not able to fulfill these glutamine-dependent reactions. These results are consistent with the proposal that GLS1 inhibition is synthetic lethal with *MYC* overexpression (Yuneva *et al*, 2012).

We next studied the effect of CDK4/6 and GLS1 inhibitors alone and in constant ratio (1:4) combinations on the viability of HCT116 colorectal cancer and MCF-7 and SK-BR-3 breast cancer cells (Fig 7A, see also Table EV4). The combination of PD0332991 and

BPTES in a wide dose range showed a strong synergistic antiproliferative effect on all three cell lines. Strong synergistic effects were also observed between PD0332991 and a second specific GLS1 inhibitor, CB-839 (Gross *et al*, 2014).

Likewise, the combination of PD0332991 and the mTOR inhibitor rapamycin exhibited a potent synergistic antiproliferative effect over a wide dose range (CI < 0.1) (Fig 7A and Table EV4). We also investigated whether the upregulation of PI3K/Akt in response to CDK4/6 inhibition represented an additional cancer cell vulnerability. Indeed, the combination of PD0332991 with the PI3K inhibitor LY294002 showed a synergistic reduction of cell proliferation over a wide dose range (Fig 7A and Table EV4), with the strongest synergistic effects achieved at low dose combinations of PD0332991 and LY294002.

None of the above drug combinations showed significant cytotoxic effects on non-cancerous BJ fibroblasts (Fig 7B; see alto

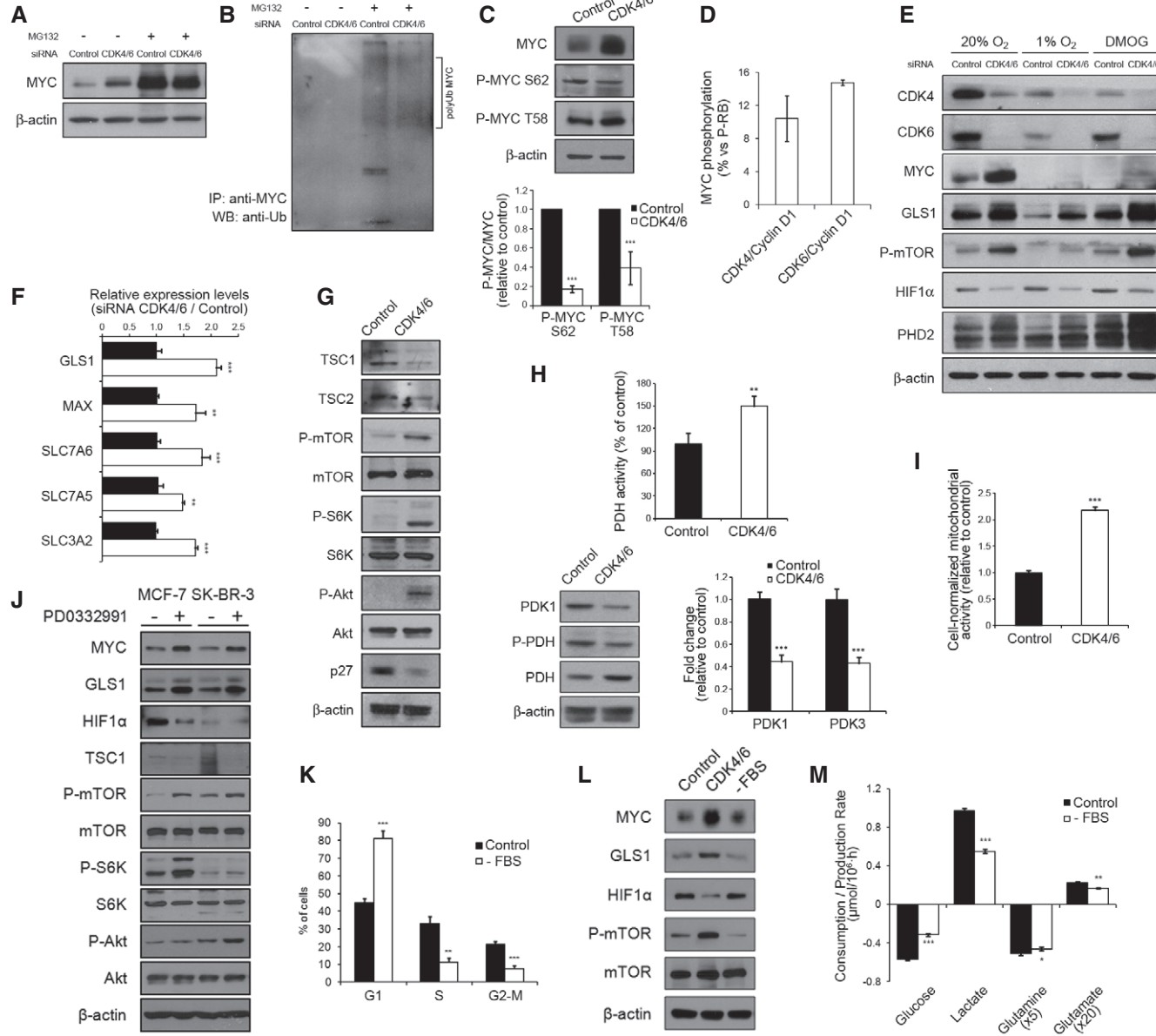

**Figure 5.**

Fig EV5A), endorsing these drug combinations as potential chemotherapeutic therapies with selective antiproliferative activity on cancer cells.

A further potential cancer cell vulnerability inferred from our analysis was a compromised adaptation to hypoxia as a consequence of CDK4/6 knockdown. Consistently, incubation for 24 h

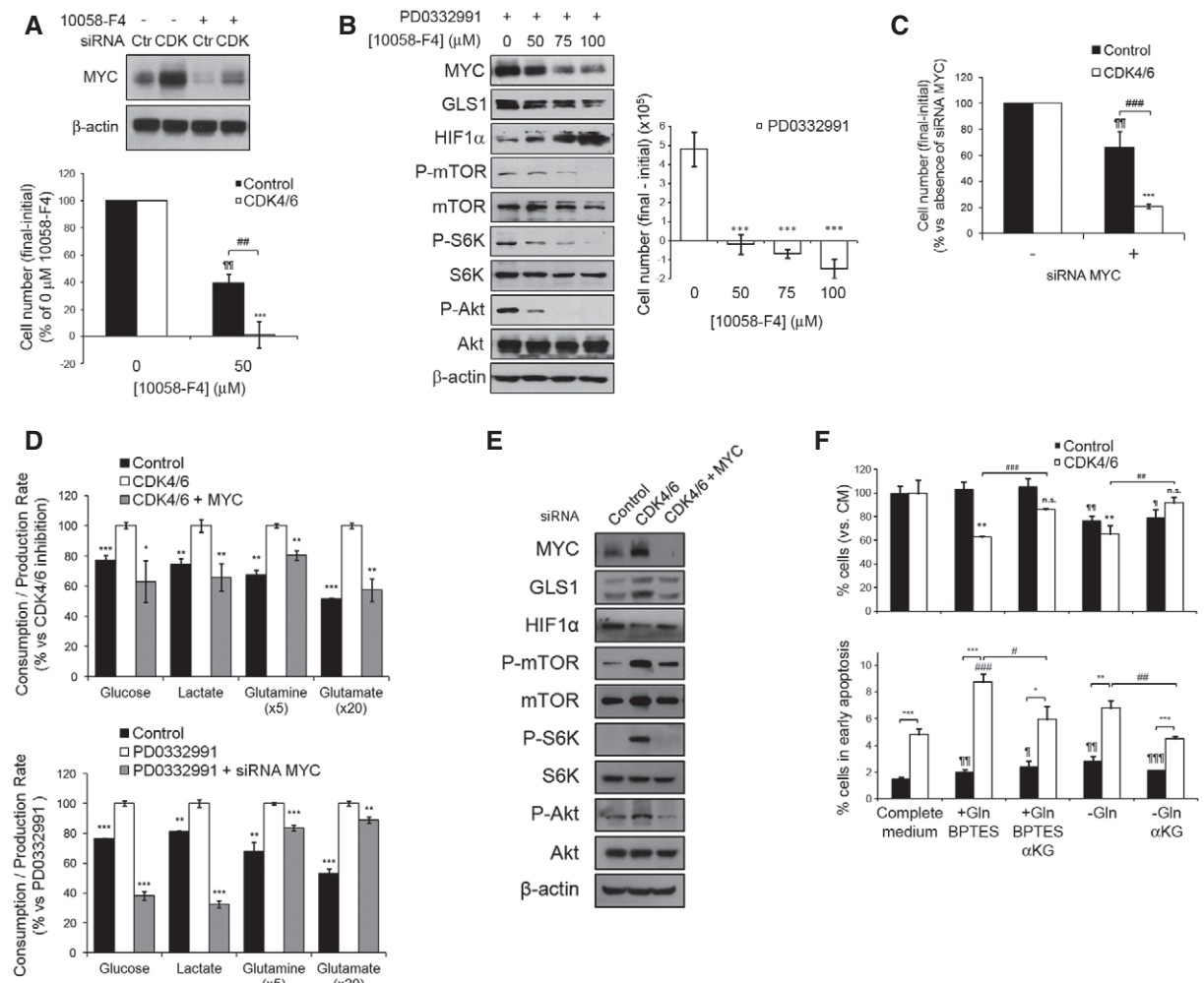

**Figure 6. CDK4/6 knockdown results in enhanced MYC dependence.**

A  CDK4/6 knockdown sensitizes HCT116 cells to MYC inhibition. Top, Western blotting analysis of MYC protein expression in CDK4/6-kd and control cells after treatment with 50 μM 10058-F4 or vehicle for 24 h. CDK, CDK4/6-kd cells; Ctr, cells transfected with non-targeting RNA duplexes. Bottom, CDK4/6-kd or control cells were scored before and after treatment with 50 μM 10058-F4 or vehicle for 24 h.

B  Inhibition of MYC reverts key molecular and signaling responses to CDK4/6 inhibition. Left, incubation of HCT116 cells with increased doses of 10058-F4 was paralleled by downregulation of MYC, GLS1, P-mTOR, P-S6K, and P-Akt and upregulation of HIF-1α. β-actin signal was used as a Western blotting loading and transfer control. Right, dose-dependent effects of 10058-F4 on the proliferation of HCT116 cells treated with PD0332991 (2 μM). Statistically significant differences are indicated as $P < 0.001$ (***).

C  MYC knockdown synergizes with CDK4/6 inhibition in its effects on HCT116 cell proliferation. Cells were counted 72 h and 96 h after siRNA transfection.

D  Effect of MYC knockdown on the extracellular metabolic fluxes of CDK4/6-kd (top) and PD0332991-treated (bottom) cells. Glucose and glutamine consumption and lactate and glutamate production rates were obtained after 24 h of incubation with fresh media and normalized to cell number. Results are expressed as percentage to CDK4/6-inhibited cells consumption and production rates.

E  Effect of CDK4/6 and MYC combined knockdown on protein levels assessed by Western blotting. Cells were transfected with MYC siRNA 24 h before analysis.

F  CDK4/6 knockdown sensitizes HCT116 cells to GLS1 inhibition. Top, Percentages of viable cells after incubation with BPTES (10 μM), BPTES (10 μM) + αKG (2 mM), under glutamine depletion or glutamine depletion + αKG (2 mM). Bottom, percentages of early apoptotic cells, assessed with the Annexin V-PI assay.

Data information: CDK4/6, CDK4/6-kd cells; Control, cells transfected with non-targeting RNA duplexes. Data are represented as mean ± SD (*n* = 3). Significance was determined by ANOVA and two-tailed independent sample Student's *t*-tests. Statistically significant differences between CDK4/6-inhibited and control cells or MYC-kd and control cells are indicated as $P < 0.05$ (*), $P < 0.01$ (**), and $P < 0.001$ (***), while differences between treatment and the corresponding control are shown as $P < 0.05$ (#), $P < 0.01$ (##), and $P < 0.001$ (###) for CDK4/6-kd and as $P < 0.05$ (¶), $P < 0.01$ (¶¶), and $P < 0.001$ (¶¶¶) for control cells. All experiments were performed 96 h after siRNA transfection or PD0332991 treatment.

Source data are available online for this figure.

under hypoxic conditions (1% $O_2$) impaired the growth of CDK4/6-kd cells while control cells were not significantly affected (Fig 7C). In addition, incubation for 24 h with 1 mM DMOG also restrained the proliferation of CDK4/6-kd cells to a greater extent than control cells (Fig 7D).

In addition to affecting growth and viability of cancer cells in 2D cultures, CDK4/6 knockdown reduced the anchorage-independent growth of HCT116 cells (Fig 7E). The combinations of PD0332991 with 10058-F4, BPTES, rapamycin, LY294002, or DMOG reduced spheroid growth significantly more than any of the drugs alone (Fig 7E; see also Fig EV5B). Remarkably, the PD0332991–BPTES combination caused the greatest and more selective effects.

# Discussion

Our metabolic and transcriptomic analyses have allowed us to precisely map the extensive metabolic reprogramming that follows CDK4/6 depletion in HCT116 colorectal cancer cells, characterized by an increase in mitochondrial metabolism and function accompanied with an enhanced metabolism of glucose, glutamine and amino acids. These observations are in agreement with those reported by Franco *et al*, who also found that CDK4/6 inhibition leads to an enhanced activation of the mTOR pathway and consequent sensitivity of pancreatic cancer cells to mTOR inhibitors (Franco *et al*, 2016). Significantly, our analysis identifies the accumulation of MYC as the major upstream event that explains most aspects of the metabolic reprogramming that follows CDK4/6 inhibition, including upregulation of mTOR. We provide evidence that MYC is a direct phosphorylation substrate of CDK4/6-Cyclin D1 complexes at Thr58 and Ser62, and depletion of CDK4/6 in HCT116 cells prevents the phosphorylation of MYC at these two residues and its proteasome-mediated degradation (Sears, 2000). As discussed below in more detail, the consequent accumulation of MYC mechanistically explains the simultaneous greater glutaminase dependence, downstream upregulation of the mTOR pathway and blunting of cellular responses to hypoxia of CDK4/6-inhibited cells.

The enhanced glucose, glutamine, and amino acid metabolism exhibited by CDK4/6-depleted cells can be explained by the ensuing upregulation of MYC. MYC enhances glycolysis through the activation of glycolytic and glucose transporters genes, promotes lactate production and export, contributes to increase glutamine uptake by upregulating the expression of glutamine transporters, and enhances glutamine metabolism by transcriptionally repressing microRNA-23a/b, resulting in a augmented expression of its target glutaminase (GLS1) (Gao *et al*, 2009; Dang, 2013). GLS1 enhances mitochondrial metabolism by catalyzing the conversion of glutamine to glutamate, a key substrate for energy and redox processes, including mitochondrial respiration, by fueling the TCA cycle through its conversion to α-KG. Consistent with our results, it has been described that pRB inhibition, whose phenotype is expected to be the opposite to CDK4/6 loss, produces a decrease on TCA cycle and mitochondrial respiration (Nicolay *et al*, 2015). Thus, our observations point to a central role played by MYC-driven glutamine metabolism in the metabolic reprogramming that follows CDK4/6 loss of function.

We have also found that CDK4/6 inhibition or knockdown is accompanied with a baseline activation of the mTOR signaling pathway, in line with prior evidences (Franco *et al*, 2016). It has been described that the mTOR-negative regulators TSC1 and TSC2 are phosphorylated and inactivated by CDK4/6, and inhibition of CDK4/6 leads to active TSC1 and TSC2 and consequent inhibition of mTOR (Zacharek *et al*, 2005). However, inhibition of CDK4/6 in our experiments leads to enhanced, not diminished, baseline activation of mTOR and thus a mechanism independent of a direct CDK4/6 regulation of mTOR must be invoked in order to explain our observations. Our study provides the following evidences in support of a MYC-driven activation of mTOR in our system: First, MYC is known to downregulate the mTOR-negative regulator TSC2 (Ravitz *et al*, 2007), as also found in our experiments. Second, the enhanced glutaminolysis that follows MYC upregulation, in turn mediated by downstream upregulation of SCL7A5 and GLS1 (Gao *et al*, 2009), would also contribute to the observed enhanced mTOR activation (Duran *et al*, 2012). In consequence, we believe that our proposal that the upregulation of MYC following CDK4/6 inhibition is a sound mechanism to explain the observed mTOR upregulation. A third major phenotype induced by inhibition of CDK4/6 in our cancer cell models is a limited response to hypoxia accompanied with downregulation of HIF-1α, which can also be explained by the accumulation of MYC. Increased levels of MYC and p-mTOR induce

---

**Figure 7. Synergistic antiproliferative effects of PD0332991 in combination with glutaminase, mTOR, or PI3K-Akt inhibitors.**

A   Synergistic antiproliferative effects between PD0332991 and BPTES, CB-839, rapamycin or LY294002. Cell viability was assessed by Hoechst staining after 96 h incubation with the combinations of inhibitors at the indicated concentrations in HCT116 (left), MCF-7 (middle), and SK-BR-3 (right) cells. Results are shown as percentage of proliferation relative to untreated cells (mean ± SD of $n = 6$).

B   The combinations of PD0332991 with BPTES, rapamycin, or LY294002 are not toxic to BJ fibroblasts. Cells were cultured at the indicated concentrations of inhibitors for 96 h, and cell proliferation was determined by Hoechst staining. Results are shown as percentage of proliferation relative to untreated cells (mean ± SD of $n = 6$).

C   CDK4/6 knockdown sensitizes cells to hypoxia. CDK4/6-kd and control cells were incubated for 24 h under normoxic or hypoxic (1% $O_2$) conditions.

D   CDK4/6-kd and control cell viability after 24-h treatment with 1 mM DMOG. Results are shown as the percentage of proliferation relative to CDK4/6-kd and control cells cultured with vehicle (100% proliferation).

E   Combinations of PD0332991 with drugs showing synergistic inhibition of spheroid growth of HCT116 cells. Left, images of HCT116 spheroids after 10 days of treatment with the indicated inhibitors. Right, quantification of total spheroid volume after treatment with the indicated drug combinations. Spheroids were scored by image acquisition followed by spheroid area and volume quantification with ImageJ. Results are shown as percentage of total spheroid volume relative to untreated cells. Data are represented as mean ± SD ($n = 5$).

Data information: CDK4/6, CDK4/6-kd cells; Control, cells transfected with non-targeting RNA duplexes. Bars represent mean ± SD ($n = 3$). Significance was determined by Kruskal-Wallis and two-tailed independent sample Student's *t*-tests. Statistically significant differences between CDK4/6-inhibited and control cells are indicated as $P < 0.01$ (\*\*) and $P < 0.001$ (\*\*\*). Differences between treatment and the corresponding control are shown as $P < 0.01$ (##) and $P < 0.001$ (###) for CDK4/6-inhibited cells and as $P < 0.01$ (¶¶) for control cells.

**Figure 7.**

glutamine transporters and GLS1, leading to both an increase of the glutamine uptake and the intracellular concentration of α-KG which, in turn, triggers the PHD2-mediated hydroxylation and subsequent degradation of HIF-1α (Tennant *et al*, 2009). This scenario provides a mechanistic explanation for the blunted adaptive responses and augmented sensitivity to hypoxia observed in CDK4/6-inhibited cells.

In sum, our metabolic and transcriptomic analyses have revealed that depletion or inhibition of CDK4/6 in cancer cells leads to *de novo* addiction to MYC, and, as likely downstream consequences, also to glutaminase and mTOR signaling, as well as to a compromised adaptation to hypoxia. These dependencies reveal vulnerabilities that can be exploited in therapeutic combinations with CDK4/6 inhibitors. This mechanism also helps explain previous observations that *CDK4* loss accelerates the development and increases the tumorigenic potential of *MYC*-driven lymphoma (Lu *et al*, 2014). Because of a lack of therapeutically efficacious MYC inhibitors (Li & Simon, 2013), we propose the inhibition of MYC downstream targets, in particular GLS1, as an effective strategy to overcome acquired tolerance and resistance of cancer cells to CDK4/6 inhibitors. This proposal is supported by the strong antiproliferative synergies found in our experiments by combining CDK4/6 knockdown or inhibition with the selective GLS1 inhibitors BPTES or CB-839. Of particular interest is the limited cytotoxic activity observed on non-malignant cells of GLS1 inhibitors as a single agent or in pairwise combinations with the other drugs tested in this study, which predicts for these combinations a significant therapeutic margin in cancer management.

# Materials and Methods

Full Materials and Methods are detailed in Appendix Supplementary Methods.

## Cell culture

Cells were obtained through the American Type Culture Collection and grown in appropriate media at 37°C. For hypoxia incubations, cells were kept in an atmosphere containing 1% oxygen and 5% $CO_2$ in a hypoxic incubator.

## siRNA transfection

ON-TARGETplus SMARTpool siRNAs directed at CDK4 (L-003238-00, GE Healthcare Dharmacon Inc., Lafayette, CO, USA) and Silencer Select siRNA against CDK6 (s51, Ambion, Austin, TX, USA) were transfected with Lipofectamine RNAiMAX (Invitrogen, Carlsbad, CA, USA). As controls, ON-TARGETplus Non-Targeting Control Pool siRNA (D-001810-10, GE Healthcare Dharmacon Inc.) and Silencer Select Negative Control siRNA (4390844, Ambion) were used.

## Cell proliferation and viability assays

Proliferation assays were performed by flow cytometry combining direct cell counting and propidium iodide staining. When testing drugs that affect mitochondrial respiration, cell viability was assessed by Hoechst staining measuring emitted fluorescence.

## Measurement of extracellular metabolites

Glucose, lactate, glutamate and glutamine concentrations were determined using a COBAS Mira Plus spectrophotometer (Horiba ABX, Kyoto, Japan). Concentrations of non-essential and essential amino acids were determined by ion-exchange chromatography.

## Transcriptomic analysis

RNA was used to produce biotinylated cRNA that was hybridized to Affymetrix GeneChip human genome U133 Plus 2.0 arrays (Affymetrix Inc., Santa Clara, CA, USA). Data were standardized using the robust multi-array average (RMA) method (Irizarry *et al*, 2003) and differential gene expression was assessed using the LIMMA package (Smyth, 2004). GSEA (Subramanian *et al*, 2005) was applied to infer gene signatures significantly associated with differentially expressed genes.

## $^{13}$C-tracer-based metabolomics

Media samples and cell extracts were used to assess the isotopologue distributions of extracellular and intracellular metabolites, respectively. Isotopologue distribution analyses of $^{13}$C-labeled metabolites were conducted by gas chromatography coupled to mass spectrometry (GC/MS).

## Oxygen consumption rate (OCR) and extracellular acidification rate (ECAR)

OCR and ECAR were determined using a XF24 Extracellular Flux Analyzer (Seahorse Bioscience, North Billerica, MA, USA).

## Data and software availability

The primary datasets and computer code produced in this study are available in the following databases: Microarray data: Gene Expression Omnibus GSE84597 (https://www.ncbi.nlm.nih.gov/geo/query/acc.cgi?acc=GSE84597). Modeling computer scripts: Zenodo database (https://doi.org/10.5281/zenodo.546717).

Expanded View for this article is available online.

## Acknowledgements

The authors thank Anibal Miranda, Dr. Ibrahim Halil Polat, Ursula Valls, and Erika Zodda for their technical support and Dr. Mary Pat Moyer for the kind gift of NCM460 cells. This work was supported by grants to M.C. from Agència de Gestió d'Ajuts Universitaris i de Recerca (AGAUR)—Generalitat de Catalunya, (2014SGR1017), ICREA Foundation (ICREA Academia) and MINECO from the Spanish Government and FEDER funds (SAF2014-56059-R, SAF2015-70270-REDT, European Commission FEDER—Una manera de hacer Europa), and to T.M.T. from MINECO (SAF2012-40017-C02-01 and SAF2015-66984-C2-1-R, European Commission FEDER—Una manera de hacer Europa) and Xarxa de Referència en Biotecnologia. This work was also supported by the Francis Crick Institute which receives its core funding from Cancer Research UK (FC001223), the UK Medical Research Council (FC001223), and the Wellcome Trust (FC001223). The CRG/UPF Proteomics Unit is part of the "Plataforma de Recursos Biomoleculares y Bioinformáticos (ProteoRed)" supported by grant PT13/0001 of ISCIII and Spanish Ministry of Economy and Competitiveness.

## Author contributions

MT-C designed and performed the experiments, analyzed and interpreted data, prepared tables and figures, and wrote the manuscript; PdA proposed and performed biocomputational analyses and pathway modeling, prepared tables and figures, and participated in manuscript organization and writing; JT-C performed experiments, analyzed and interpreted data, and participated in manuscript organization and writing; JP performed experiments and analyzed and interpreted data; MY prepared and supplied key reagents, proposed, and interpreted experiments; TMT performed data analysis, proposed and interpreted experiments, and participated in manuscript organization and writing; MC built the research team, obtained funds, performed, supervised, interpreted the experimental design and analysis, and participated in manuscript organization and writing.

## Conflict of interest

The authors declare that they have no conflict of interests.

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
