## [Review Process File · Molecular Systems Biology]

De novo MYC addiction as an adaptive response of cancer cells to CDK4/6 inhibition

Míriam Tarrado-Castellarnau, Pedro de Atauri, Josep Tarragó-Celada, Jordi Perarnau, Mariia Yuneva, Timothy M. Thomson & Marta Cascante

Corresponding author: Marta Cascante, University of Barcelona

Review timeline:

Submission date:	16 September 2016
Editorial Decision:	08 November 2016
Revision received:	21 April 2017
Editorial Decision:	06 July 2017
Revision received:	03 August 2017
Accepted:	15 August 2017

Editor: Thomas Lemberger

Transaction Report:

1st Editorial Decision

08 November 2016

Thank you again for submitting your work to Molecular Systems Biology. We have now heard back from the three referees who agreed to evaluate your manuscript. As you will see from the reports below, the referees find the topic of your study of potential interest. They raise, however, several concerns, which we would ask you to address convincingly in a major revision.

Without repeating all the points mentioned by the reviewers in their reports, the key elements are the following:

- a number of additional experiments appear to be required to ascertain the role of MYC: the impact of a direct knock down of MYC (referee #1) should be performed and the direct phosphorylation of MYC by CDK4/6 (referee #3) should be demonstrated and characterized. This seems essential as the major aspect of novelty of the present work is to show that the metabolic alterations are orchestrated by MYC.

- a second decisive point is to understand whether the observed alterations are caused by CDK4/6 inhibition or rather reflect the consequences of cell cycle arrest. Both reviewer #1 and #3 suggest experiments to characterize the effect of CDK4/6 inhibitors and compare their effects to comparable (G0/G1) cell cycle arrest.

- the rigour of the flux analysis should be considerably improved and an accurate report of the methods used should be reported. The computer scripts used in this study should be provided in a form that allows others to understand how to use them and to reproduce the results. Scripts can be uploaded as zip archives with a README file at the top level of the folder and called out from the text as "Computer code".

 REVIEWER REPORTS

Reviewer #1:

The manuscript by Tarrado-Castellamau reports the intriguing induction of MYC protein upon CDK4/6 inhibition. Increased MYC was associated with increased glycolysis, glutaminolysis, mitochondrial function and oxygen consumption, but fatty acid synthesis was diminished in HCT116. The authors performed flux measurements using ¹³C-labeled substrate. The authors infer that knockdown or inhibition of CDK4/6 resulted in diminished direct phosphorylation of MYC (Ser 62) resulting in less MYC degradation. The authors further show that amino acid transporters were increased with inhibiting CDK4/6 that also induced mTOR activity and repressed TSC2. The authors further documented that CDK4/6 inhibition sensitized cells to inhibition of MYC, GLS, mTOR, or PI3K.

1) Although the authors claim that MYC inhibition synergizes with CDK4/6 inhibition, this was not definitively demonstrated. Since the MYC-MAX dimerization inhibitor may have off-target effects, direct knock-down of MYC would add greatly to the evidences. The authors should address the activity of 10058-F4 that should disrupt MYC-MAX dimerization, but Figure 6B show a decrease in MYC protein level; this phenomenon should be discussed particularly since the authors did not demonstrate the disruption of MYC and MAX dimerization directly.

2) A fundamental aspect of cell growth in response to CDK4/6 should be addressed in this work. What happen to cell growth (cell size), cell proliferation (cell number), and cell cycle distribution when cells are treated with CDK4i/6i? Do cells stop entering S-phase, but continue to accumulate cellular mass? Note that CDK4 is a critical MYC target; so it is possible that MYC-induced cell mass accumulation cause cells to be dependent on mTOR. The use of thymidine block in Figure 5J in HCT116 is an inadequate approach to rule out the role of the cell cycle as stated by the authors on Page 13: "These observations support the conclusion that the modulation of MYC, GLS1, PmTOR and HIF-1 α that follows CDK4/6 depletion or inhibition is not a result of G1 arrest but directly attributable to a loss of function of CDK4/6. This would potentially provide an understanding of synthetic lethal interaction between CDK4/6 and mTOR inhibition." The authors should directly provide flow cytometric measurements of the cell cycle effect of PD0332991 as well as cell size (an FSC vs SSC scatter flow cytometric analysis should be shown). I am unconvinced by the statement made on Page 13.

It is particularly important to understand why some of the other selected pathways could be targeted for synthetic lethality in similar manner. If cells stop in the cell cycle with CDK4i/6i treatment, then there is a conceptual framework that is missing in the current manuscript. If CDK4i/6i blocks cells from entering S-phase, then what is the effect on cell growth (mass) and why do some of these other pathways open up vulnerabilities for the kinase inhibitor treated cells. As it stands, this manuscript reports a number of intriguing findings, but the mechanistic insight could be provided with further depth. Synergies between inhibitors are indeed very interesting; however, how these synergies fit with decreased CDK4/6 is unclear. For example, why are cells dependent on PI3K signaling with CDK4i/6i-induced MYC expression?

3) The statement on Page 12 (last line): "Thus, treatment of MCF-7 or SKBR-3 breast cancer cells with the CDK4/6 inhibitor PD0332991 also caused an upregulation of MYC, GLS1 and the mTOR and Akt signaling pathways, as well as a downregulation of HIF-1 α " does not seem to be supported by Figure 5I. Note that GLS level was relatively high in SK-BR-3 and was not further induced by PD0332991; so the statement should be refined.

Reviewer #2:

The manuscript describes an analysis of the metabolic and transcriptional state of cancer cells that

rely on CDK4/6 - CycD1 for cell cycle regulation. This is a peculiar feature of some tumors and an ideal target for selective inhibition. This study focused on the metabolic consequence of depleting CDK4 and 6, in an attempt to find synergistic targets for specific and cytotoxic intervention. The paper is well written. It develops mostly linearly with a clear logic and sound experiments. The key claims are neatly supported by experiments. Eventually, the presented experiments reveal that - upon CDK4/6 depletion - resilient cells acquire a dependency on MYC and, in turn, mTOR and GLS1 to fuel the TCA cycle with carbon from glutamine. Drugs inhibiting these processes exhibit the strongest inhibition in combination with CDK4/6 KD or pharmacological inhibition. The phenotype was verified in 3D models.

I have two major concerns.

(1) The biggest problem is novelty. This study was (unfortunately) scooped by the Franco et al Cell Reports 2016 paper. The Franco et al study already reported identical (i) increase in metabolic consumption of glucose, glutamine, and oxygen upon CDK4/6 inhibition in p16INK4a^{-/-} cells, (ii) increase in mitochondrial activity (and mass), and (iii) synthetic lethality with mTOR.

This novel submission reports the same results. Additionally, it highlights the role of MYC stabilization upstream of mTOR, and the metabolic dependence on glutamine and glutaminase to replenish the TCA cycle. These are the key novel insights. They were inspired by a transcriptome and gene set enrichment analysis, and validated by Western Blots.

One additional novel aspect of this submission is the detailed metabolic characterization of CDK4/6-KD cells presented in the first half of the text. Regardless of specific concerns on the correctness of the metabolic results (more below), the relevance of the metabolic analysis part is overblown. The magnitude of all TCA flux changes are minor, and fail to explain or simply hypothesize what causes the dependency on glutamine. Albeit very detailed, the analysis is merely descriptive. Notably, complete removal of the labeling and derived flux data (i.e. Figure 2) wouldn't harm the story.

Overall, the relevance of network biology is marginal. Novel insights are derived from traditional reasoning and experiments.

(2) The metabolic analysis is qualitatively weak and likely biased. Over the past years, I learned that flux calculation for cells is very difficult if not impossible. However, figure 2 shows spectacular results: precise calculation of fluxes in different compartments. This is by far more detailed than anything published thus far. I had a look at the flux calculation method, and it seems that the authors came up with a new approach which apparently requires plenty of tweaking. The authors use terms such as "playing with the [...] scripts" and "playing with the ratios", which let me think that the fluxes were obtained with substantial manual curation. I glanced through the estimated labeling patterns (Supp Table 1), and several don't seem to fit with the measured values (for example lactate, $\delta > 10\%$!). Surprisingly, there is no measure of the goodness of the fit. These are all unacceptable practices because they introduce massive bias and overfitting.

For transparency and formal correctness, the authors should use validated software (Metran, INCA), provide a formal estimation of confidence intervals, and provide the files necessary to reproduce the analysis.

Minor points

(3) A minor point is the controversial result on HIF1 α , which is lower in CDK4/6-KD cells. Isn't this in conflict with the observation that a notorious target of HIF1 α such as glycolysis is actually higher?

(4) Is dependency on MYC in CDK4/6-KD cells conserved in cells that lost Cyclin E or RB?

Reviewer #3:

The manuscript by Tarrado-Castellamau et al. reports a metabolic reprogramming induced by Cdk4/6 inhibition. In particular, CDK4/6 knockdown results in increased glycolysis accompanied by enhanced glutamine oxidation and contribution to the oxidative TCA cycle, as well as increased

concentration of TCA intermediates. Glutamine seems the preferred substrate for respiration. Interestingly, these changes are associated to MYC accumulation and a MYC-dependent transcriptional response leading to increased glutamine metabolism, mTOR activation and HIF1 α -mediated responses. As a result, these cells become sensitive to inhibitors of these pathways, suggesting new combinatory approaches to improve the use of CDK4/6 inhibitors in the clinic. In general, the manuscript describes a number of very interesting observations regarding the metabolic changes induced by CDK4/6 inhibition using either RNAi or palbociclib, a CDK4/6 specific inhibitor recently approved for treating hormone-positive breast cancer. The quality of the metabolic analysis is very high and, in general, this work provides the most complete analysis of the metabolic changes induced by CDK4/6 inhibition so far.

On the other hand, the quality of the cell biology analysis is not as high as the metabolic part and the authors should improve some of the aspects of the manuscript as describe below.

Major points

1. Role of MYC in the phenotypes observed. The authors attribute to MYC accumulation most of the metabolic changes: see for instance "by identifying the accumulation of MYC as the major upstream event that explains most aspects of the metabolic reprogramming that follows CDK4/6 inhibition." (pag. 16) or "depletion or inhibition of CDK4/6 in cancer cells leads to de novo addiction to MYC" (page 17). However, the role of MYC as a major CDK4/6 target is not demonstrated. The early evidences that MYC peptides are CDK4 substrates (Anders et al., 2011) have very limited value and the relevance of a possible phosphorylation of MYC by CDK4/6 has not been explored in the past. Similarly, the exact phosphorylation site has not been analyzed and it is not clear why the authors use pSer62 as a read-out of CDK4/6 activity. In general, our knowledge on the relevance of MYC phosphorylation by CDK4/6 is too limited to drawn any conclusion. There are many "indirect" reasons why MYC may be less phosphorylated apart from direct phosphorylation by CDK4/6. In addition, the effect of MG132 is much more obvious than the effect of CDK4/6 inhibition and MG132 would lead to a similar effect in any protein degraded in a proteasome-dependent manner without indicating a direct effect. Thus, a more detailed analysis of the MYC-dependent phosphorylation by CDK4/6 is required for supporting the major conclusion in the manuscript. A direct manner to demonstrate the role of MYC would be to identifying MYC phosphorylation site and expressing phosphomimetic/phosphoresistant mutants to rescue/mimic the phenotypes induced by CDK4/6 inhibition.

2. Both glycolysis (ECAR) and oxidative respiration (OCR) seem to be increased upon CDK4/6 knockdown. It seems that CDK4/6 inhibition may trigger a general energetic-stress response rather than specifically modulate glycolysis. Thus, one of the most important concerns in the manuscript is to what extent the observations are caused by direct CDK4/6 knock-down/inhibition or by cell cycle arrest. To address this question the authors use thymidine-arrested cells. This is not the right control. Thymidine treatment results in defective S-phase and in an early S-phase arrest, whereas inhibition of CDK4/6 should arrest cells in G0-early/mid G1. This is a crucial point as all the phenotypes observed may be a consequence of G0/G1 arrest. The authors should compare CDK4/6 inhibition with a pure G0 arrest, for instance, serum deprivation versus CDK4/6 inhibition in non-transformed cells.

3. Similarly, the characterization of the TOR response to CDK4/6 inhibition is limited to discussion of published data. CDK4/6 are known to directly phosphorylate TSC2 and this may explain some phenotypes presented in the manuscript. Same rationale applies to P-Akt, a typical mark downstream of several feedback loops downstream of TOR. In general, it is not clear to what extent the Akt-TSC2-TOR is modified as a direct target or a secondary consequence of the alterations induced by CDK4/6 knockdown/inhibition. Similarly, It is very difficult in the manuscript to conclude on to what extent the upregulation of MYC is responsible for the changes in TOR and metabolic pathways in the cell. Or similarly, whether the changes in TOR are responsible for the metabolic changes.

Other points

Lack of p16 does not automatically results in "constitutive activation of CDK4/6" (page 4). These kinases still require binding to the cyclin, activating phosphorylation, etc.

CDK4/6 knockdown results in increased apoptosis (Fig. 1C). This is somehow unexpected for

CDK4/6 knockdown. Any hypothesis for this? Are the authors sure about the specificity of the RNAi reagents?

Not sure if the effects in Figure 7 represent synergy or cooperation. This should be tested and discussed appropriately.

Why CDK4/6 knockdown reduces the anchorage-independent growth of HCT116 cells? A role of CDK4/6 in anchorage-independent growth has not been proposed in the past.

Fig. 1M uses palbociclib whereas the rest of the data have been generated with CDK4/6 knock-down.

Figure 1B (S1A). Does "proliferation" mean number of cells? The definition of these variables is not clear in the figures. For instance in Fig. S1C the authors indicate "survival" in the y axis but describe proliferation in the figure legend.

Figure 1D (S1B). FACS (PI) profiles do not indicate percentage of G1 or G2/M cells but DNA content (2n, 4n, etc)

Fig. S1A could be combined with Fig. 1B.

The authors may want to discuss recent papers (see for instance papers by E. Knudsen or N. Dyson) on the effect of palbociclib (or Cdk4-pRb) in mitochondrial function.

1st Revision - authors' response

21 April 2017

Responses to Reviewer #1:

The manuscript by Tarrado-Castellarnau reports the intriguing induction of MYC protein upon CDK4/6 inhibition. Increased MYC was associated with increased glycolysis, glutaminolysis, mitochondrial function and oxygen consumption, but fatty acid synthesis was diminished in HCT116. The authors performed flux measurements using ¹³C-labeled substrate. The authors infer that knockdown or inhibition of CDK4/6 resulted in diminished direct phosphorylation of MYC (Ser 62) resulting in less MYC degradation. The authors further show that amino acid transporters were increased with inhibiting CDK4/6 that also induced mTOR activity and repressed TSC2. The authors further documented that CDK4/6 inhibition sensitized cells to inhibition of MYC, GLS, mTOR, or PI3K.

1) Although the authors claim that MYC inhibition synergizes with CDK4/6 inhibition, this was not definitively demonstrated. Since the MYC-MAX dimerization inhibitor may have off-target effects, direct knock-down of MYC would add greatly to the evidences. The authors should address the activity of 10058-F4 that should disrupt MYC-MAX dimerization, but Figure 6B show a decrease in MYC protein level; this phenomenon should be discussed particularly since the authors did not demonstrate the disruption of MYC and MAX dimerization directly.

We thank the reviewer for raising this issue. The molecule used in these experiments, 10058-F4, has been demonstrated to disrupt the MYC-MAX complex in previous studies (Yin et al., 2003). It has also been reported that MYC levels are downregulated in cells treated with this molecule, presumably due to diminished stability of monomeric MYC relative to its heterodimeric forms (Huang et al., 2006; Lin et al., 2007; Zirath et al., 2013). We have included the following reference to this in the revised manuscript (page 14, paragraph 2):

In agreement with previous studies (Lin et al., 2007), treatment with the MYC-MAX heterodimerization inhibitor 10058-F4 (Yin et al., 2003) strongly downregulated MYC protein levels causing a 60% decrease in control cell numbers and a complete abrogation of proliferation in CDK4/6-kd cells at 50 μM (Figure 6A). In addition, 10058-F4 treatment caused a dose-dependent inhibition of cell proliferation and a reversal of the effects of PD0332991 on GLS1, HIF-1α, P-mTOR, P-S6K and P-Akt protein levels (Figure 6B).

Moreover, in order to further confirm the dependence on MYC of CDK4/6-inhibited HCT116 cells, we have conducted experiments in which MYC was knocked down with specific RNAi duplexes. The results of these experiments, shown in the new Figure 6C-E, indicate that, similar to treatment with 10058-F4, MYC knockdown also synergizes with CDK4/6 inhibition in its effects on HCT116 cell proliferation, and reinforce our conclusion that a major adaptive response by cancer cells to CDK4/6 inhibition is MYC addiction. Likewise, we now show that MYC knockdown in CDK4/6-kd cells counteracted the effects of CDK4/6 inhibition on GLS1, HIF-1 α , P-mTOR, P-S6K and P-Akt protein levels. Furthermore, we also found that MYC knockdown reduced the consumption rate of glucose and glutamine and the production rate of lactate and glutamate in both CDK4/6-kd and PD0332991-treated cells. We have added this sentence in the results section (page 14, paragraph 2):

Likewise, specific knockdown of MYC with RNAi duplexes had the same effect as 10058-F4 treatment to synergize with CDK4/6 knockdown in reducing cell proliferation (Figure 6C) and to reverse the effects of CDK4/6 inhibition both on the consumption and production rates of glucose, glutamine, lactate and glutamate (Figure 6D) and on GLS1, HIF-1 α , P-mTOR, P-S6K and P-Akt protein levels (Figure 6E).

2) A fundamental aspect of cell growth in response to CDK4/6 should be addressed in this work. What happens to cell growth (cell size), cell proliferation (cell number), and cell cycle distribution when cells are treated with CDK4i/6i? Do cells stop entering S-phase, but continue to accumulate cellular mass? Note that CDK4 is a critical MYC target; so it is possible that MYC-induced cell mass accumulation causes cells to be dependent on mTOR. The use of thymidine block in Figure 5J in HCT116 is an inadequate approach to rule out the role of the cell cycle as stated by the authors on Page 13: "These observations support the conclusion that the modulation of MYC, GLS1, P-mTOR and HIF-1 α that follows CDK4/6 depletion or inhibition is not a result of G1 arrest but directly attributable to a loss of function of CDK4/6. This would potentially provide an understanding of synthetic lethal interaction between CDK4/6 and mTOR inhibition." The authors should directly provide flow cytometric measurements of the cell cycle effect of PD0332991 as well as cell size (an FSC vs SSC scatter flow cytometric analysis should be shown). I am unconvinced by the statement made on Page 13.

It is particularly important to understand why some of the other selected pathways could be targeted for synthetic lethality in similar manner. If cells stop in the cell cycle with CDK4i/6i treatment, then there is a conceptual framework that is missing in the current manuscript. If CDK4i/6i blocks cells from entering S-phase, then what is the effect on cell growth (mass) and why do some of these other pathways open up vulnerabilities for the kinase inhibitor treated cells. As it stands, this manuscript reports a number of intriguing findings, but the mechanistic insight could be provided with further depth. Synergies between inhibitors are indeed very interesting; however, how these synergies fit with decreased CDK4/6 is unclear. For example, why are cells dependent on PI3K signaling with CDK4i/6i-induced MYC expression?

Following the reviewer's indications, we have included in our revised manuscript new graphs illustrating FS/SS, cell volume, DNA content and cell cycle analyses after treatment of HCT116 cells with CDK4/6-specific RNAi or with PD0332991 (Figure EV1 in the revised manuscript). These experiments illustrate the extent of G1 arrest as a consequence of either treatment, as well as an increased average cell volume. We have included these observations in the results section (page 5, paragraph 1):

CDK4/6 inhibition caused an increase in cell volume but had no effect on the total cellular protein content (Figure EV1A-D).

In order to test whether the observed changes were a result of the accumulation of cells in the G1 cell cycle phase, we performed experiments with serum deprivation on HCT116 human colorectal carcinoma cells and non-transformed NCM460 human epithelial cells derived from healthy colon mucosa (Moyer et al., 1996), to obtain a G0/G1 cell cycle arrest. We found that serum deprivation induced a greater G0/G1 arrest than CDK4/6 inhibition, as seen in Figure 1D and Figure EV1G compared to Figure 5K for HCT116 cells and in Figure EV4G for NCM460 cells. Our results showed that in both HCT116 and NCM460 cell lines, cells synchronized in the G1 phase of the cell cycle presented comparable protein levels of MYC, GLS1, P-mTOR and HIF-1 α to asynchronous cells. We also measured the consumption and production rates of glucose, glutamine, lactate and

glutamate in serum-deprived and control cells. HCT116 G0/G1 synchronized cells presented a statistically significant decrease of all these rates while NCM460 serum-starved cells did not change their consumption and production rates compared to control cells. Collectively, these results support the conclusion that the modulation of MYC, GLS1, P-mTOR and HIF-1 α that follows CDK4/6 depletion or inhibition is not a result of G1 arrest but directly attributable to a loss of function of CDK4/6. We have included these findings in the results section (page 13, paragraph 3):

Cells synchronized in the G1 phase of the cell cycle (Figure 5K) did not accumulate MYC, GLS1 or P-mTOR as compared to asynchronous cells, and presented higher HIF-1 α levels than cells with CDK4/6 inhibition (Figure 5L). We found the same results with NCM460 human epithelial cells derived from healthy colon mucosa (Figure EV4G-H). Further, the consumption and production rates of glucose, glutamine, lactate and glutamate in G1-synchronized cells were significantly lower than those of control HCT116 cells (Figure 5M) and not significantly different from asynchronous NCM460 cells (Figure EV4I). These observations support the conclusion that the modulation of MYC, GLS1, P-mTOR and HIF-1 α that follows CDK4/6 depletion or inhibition is not a result of G1 arrest but directly attributable to a loss of function of CDK4/6.

With regards to the possibility that the increase in cell mass, rather than MYC upregulation, may underlie the increased mTOR dependence of cells inhibited for CDK4/6, we present the following argument: CDK4/6 inhibition, in its own right, caused accumulation of cells in G1 and a concomitant increase in cell size (Figure EV1). In parallel experiments, cells that were blocked in early S by double thymidine block also underwent a significant increase in average cell size (see figure below). Yet the latter cells failed to display increased baseline mTOR activation as assessed by Western blotting with anti-phospho-mTOR antibodies:

(Figures for referees not shown)

Thus, it is likely that the activation of mTOR observed in our experiments is not merely a consequence of increased cell size, and alternative mechanisms need to be invoked. First, MYC is known to downregulate the mTOR negative regulator TSC2 (Ravitz et al., 2007), as also found in our experiments (Figure 5G). Second, the enhanced glutaminolysis that follows MYC upregulation would also contribute to the observed enhanced mTOR activation (Duran et al., 2012). In consequence, we believe that our proposal that the upregulation of MYC following CDK4/6 inhibition is a sound mechanism to explain the observed mTOR upregulation. Anti-proliferative synergies between PI3K inhibition and CDK4/6 inhibition have previously been reported and attributed to a downregulation of cyclin D1 levels combined with diminished RB phosphorylation upon inhibition of PI3K-Akt signaling (Jansen et al., 2017; Vora et al., 2014). The upregulation of mTOR activity upon CDK4/6 inhibition, shown in our study, is expected to lead to activation of TORC2, which would explain the enhanced phosphorylation of Akt on Ser473 observed in our experiments. Phosphorylation at the C-terminal domain of Akt (including Ser473) provides enhanced stability to the protein and thus permits full activation of the enzymatic activity of Akt upon phosphorylation of its activation loop site (including Thr308). Phosphorylation of Thr308 and other activation loop residues is mediated by PDK, in turn activated by PI3K and upstream signaling (reviewed in (Risso et al., 2015)). In our experiments, inhibition of PI3K (with LY294002) would result in a loss of phospho-Thr308 and consequent diminished Akt activity, even when retaining phospho-Ser473 (through TORC2 activation).

3) The statement on Page 12 (last line): "Thus, treatment of MCF-7 or SKBR-3 breast cancer cells with the CDK4/6 inhibitor PD0332991 also caused an upregulation of MYC, GLS1 and the mTOR and Akt signaling pathways, as well as a downregulation of HIF-1 α " does not seem to be supported by Figure 5I. Note that GLS level was relatively high in SK-BR-3 and was not further induced by PD0332991; so the statement should be refined.

We thank the reviewer for bringing this to our attention. We agree that the quality of the Western blotting image does not allow to ascertain whether there is a significant difference between the GLS1 protein levels with or without PD0332991 treatment in SK-BR-3 cells. In response we have

repeated the corresponding experiments in order to obtain a more unambiguous Western blotting image, which supports that PD0332991 caused an upregulation of GLS1 protein levels (Figure 5J).

Responses to Reviewer #2:

The manuscript describes an analysis of the metabolic and transcriptional state of cancer cells that rely on CDK4/6 - CycD1 for cell cycle regulation. This is a peculiar feature of some tumors and an ideal target for selective inhibition. This study focused on the metabolic consequence of depleting CDK4 and 6, in an attempt to find synergistic targets for specific and cytotoxic intervention. The paper is well written. It develops mostly linearly with a clear logic and sound experiments. The key claims are neatly supported by experiments. Eventually, the presented experiments reveal that - upon CDK4/6 depletion - resilient cells acquire a dependency on MYC and, in turn, mTOR and GLS1 to fuel the TCA cycle with carbon from glutamine. Drugs inhibiting these processes exhibit the strongest inhibition in combination with CDK4/6 KD or pharmacological inhibition. The phenotype was verified in 3D models.

I have two major concerns.

1. The biggest problem is novelty. This study was (unfortunately) scooped by the Franco et al Cell Reports 2016 paper. The Franco et al study already reported identical (i) increase in metabolic consumption of glucose, glutamine, and oxygen upon CDK4/6 inhibition in p16INK4a^{-/-} cells, (ii) increase in mitochondrial activity (and mass), and (iii) synthetic lethality with mTOR.

This novel submission reports the same results. Additionally, it highlights the role of MYC stabilization upstream of mTOR, and the metabolic dependence on glutamine and glutaminase to replenish the TCA cycle. These are the key novel insights. They were inspired by a transcriptome and gene set enrichment analysis, and validated by Western Blots.

We respectfully disagree with the reviewer's viewpoint that our report is conceptually equivalent to that of Franco *et al.* There are clearly overlapping findings with regards to mTOR activation in response to CDK4/6 inhibition. However, the most significant discovery described in our study is precisely that CDK4/6 inhibition entails stabilization of MYC, which largely explains the metabolic and phenotypic adaptations to CDK4/6 inhibition. In addition to evidences provided in our original submission, we now provide evidence that CDK4/6 directly phosphorylates MYC on Thr58 and Ser62 to support our conclusions that MYC is a direct target of CDK4/6. We argue that this is a novel mechanism explaining the adaptation of cells to CDK4/6 inhibition, a mechanism that has not been addressed or noted previously, including Franco *et al.* This new mechanism is not solely based on transcriptomic and Western blotting analysis:

(1) Our metabolic analysis strongly suggests a MYC-dependent metabolic reprogramming upon CDK4/6 inhibition.

(2) We provide strong functional evidences in support of the *de novo* MYC addition hypothesis by means of MYC knockdown, pharmacological inhibition of MYC-MAX heterodimers and of MYC downstream targets. We believe that our discovery is not a minor incremental advancement over prior knowledge, but rather a major new conceptual advancement, in that we have unveiled a novel mechanism used by cancer cells for acquired drug tolerance and/or resistance: a critical switch in cellular addictions to oncogenic signals triggered by targeted therapies, which entails a critical switch in vulnerabilities.

2. One additional novel aspect of this submission is the detailed metabolic characterization of CDK4/6-KD cells presented in the first half of the text. Regardless of specific concerns on the correctness of the metabolic results (more below), the relevance of the metabolic analysis part is overblown. The magnitude of all TCA flux changes are minor, and fail to explain or simply hypothesize what causes the dependency on glutamine. Albeit very detailed, the analysis is merely descriptive. Notably, complete removal of the labeling and derived flux data (i.e. Figure 2) wouldn't harm the story.

Overall, the relevance of network biology is marginal. Novel insights are derived from traditional reasoning and experiments.

We understand the concerns expressed by the referee regarding the magnitude of the changes affecting the fluxes through the TCA cycle. The relevance of changes in fluxes can be assessed by observing changes in metabolic intermediates, which are a function of the relative changes in metabolite supply and demand capacities. A change in the relative supply and demand maximal rates for a given metabolite that causes large fluctuations in the concentrations of such a metabolite

may be accompanied with relatively small changes in the corresponding reaction fluxes. For instance, the measured increase in OCR by a factor of 1.3 that follows CDK4/6 inhibition would represent a significant increase in the production of ATP and ROS. Indeed, a significant increase in the levels of ROS was observed as well as in NADPH, α -ketoglutarate or glutamate. In our opinion, the flux reprogramming mapped in Figure 3 provides important keys to evaluate the increased glutamine dependency of CDK4/6-inhibited cells. The estimated decrease in the flux through malate dehydrogenase is very significant if compared with the increases in glycolysis and mitochondrial respiration. Upon CDK4/6 inhibition, the decreased flux through the cytosolic malate dehydrogenase (MDH) implies a reduced transport of reducing equivalents from glycolytic NADH into mitochondria to fuel the mitochondrial respiratory chain. Also, the initiation of the Krebs cycle by pyruvate dehydrogenase is not significantly perturbed under these conditions. In contrast, the flux map illustrates how the increased glutamine uptake and transformation to glutamate by glutaminase fuels mitochondrial respiratory activity. Altogether, this provides a mechanism to explain a higher dependency on glutamine and lower dependency on glucose in response to CDK4/6 inhibition. We have added the following paragraph (page 8, paragraph 1):

The outcome of this analysis was compatible with all prior observations, supporting that CDK4/6-kd cells exhibit a higher mitochondrial activity accompanied with a higher dependence on glutamine and lower dependence on glucose (Figure 3 and Table EV1). Accordingly, the decreased flux through the cytosolic malate dehydrogenase (MDH) implies a reduced transport of reducing equivalents from glycolytic NADH into mitochondria to fuel the mitochondrial respiratory chain. In addition, the increased glutamine uptake and transformation to glutamate by glutaminase (GLS1) also promotes the mitochondrial respiratory activity since glutamate is the substrate for mitochondrial glutamate dehydrogenase (GDH).

3. The metabolic analysis is qualitatively weak and likely biased. Over the past years, I learned that flux calculation for cells is very difficult if not impossible. However, figure 2 shows spectacular results: precise calculation of fluxes in different compartments. This is by far more detailed than anything published thus far. I had a look at the flux calculation method, and it seems that the authors came up with a new approach which apparently requires plenty of tweaking. The authors use terms such as "playing with the [...] scripts" and "playing with the ratios", which let me think that the fluxes were obtained with substantial manual curation. I glanced through the estimated labeling patterns (Supp Table 1), and several don't seem to fit with the measured values (for example lactate, $\delta > 10\%$!). Surprisingly, there is no measure the goodness of the fit. These are all unacceptable practices because they introduce massive bias and overfitting.

For transparency and formal correctness, the authors should use validated software (Metran, INCA), provide a formal estimation of confidence intervals, and provide the files necessary to reproduce the analysis.

We appreciate the referee's recommendations. Accordingly, we have performed a completely new analysis, independent of the analysis described in our original manuscript, by using the MATLAB-based software package INCA as follows:

- (1) A parameter optimization process was applied using INCA, including the generation of statistical metrics used to assess the goodness of the fit.
- (2) Confidence intervals of the entire flux map were estimated to convey the uncertainties associated with all estimated parameters.
- (3) Some reactions, such as those associated with the metabolism of polyamines and methionine for which we had generated insufficient data, were removed from the original model.

All parameters generated by INCA, predicted values for best fit obtained by parameter optimization, the corresponding metrics to assess the goodness of the fit and confidence intervals are included in the new Expanded View Table 1, together with a detailed description of the model. Also, we provide files generated by INCA with the estimated flux map distributions, network model and tracer simulations, which can be used to reproduce the calculations (<https://doi.org/10.5281/zenodo.546717>). The outcome of this analysis is compatible with all experimental observations, supporting that CDK4/6-kd cells exhibit a higher mitochondrial activity. The new Figure 3 has been streamlined to include only processes that were modeled in our new analysis.

Minor points

4. A minor point is the controversial result on HIF1 α , which is lower in CDK4/6-KD cells. Isn't this in conflict with the observation that a notorious target of HIF1 α such as glycolysis is actually higher?

As pointed out by the reviewer, high HIF-1 α levels (such as those induced by hypoxia) upregulate the glycolytic pathway. However, other signals and factors can upregulate this pathway in the absence of high HIF-1 α protein levels, and high glycolysis does not necessarily cause increased HIF-1 α levels. Conversely, low HIF-1 α levels do not counter enhanced glycolysis triggered by independent mechanisms. Our observations fit the following mechanistic model: upregulation of MYC causes enhanced glutaminolysis which leads to higher concentrations of α -KG which in turn promotes PHD activity, leading to HIF-1 α recognition by VHL, polyubiquitination and proteasome-dependent degradation (Tennant et al., 2009). Enhanced glycolysis in this context would be explained by MYC-dependent upregulation of pro-glycolytic enzymes and transporters, and would be uncoupled from HIF-1 α levels. The reciprocal regulation of MYC and HIF-1 α has been shown to occur in several scenarios (Koshiji et al., 2004; Koshiji et al., 2005; Zhang et al., 2007). Finally, and in further agreement with our findings, prior evidence indicates that cells with MYC overexpression display increased sensitization to hypoxia due to a limited HIF-1 α transcriptional response to low oxygen tension (Brunelle et al., 2004).

5. Is dependency on MYC in CDK4/6-KD cells conserved in cells that lost Cyclin E or RB?

The cells tested in our experiments (HCT116 colorectal, MCF-7 and SK-BR-3 mammary cancer cells) lack RB and Cyclin E genomic alterations (www.cbioportal.org), and thus are expected to be proficient in both activities. This explains our experimental observation that CDK4/6 inhibition (RNAi or PD0332991) causes an accumulation of cells in G1. Cells with amplified Cyclin E or loss of RB1 are resistant to CDK4/6 inhibition (Herrera-Abreu et al., 2016; Konecny et al., 2011; Logan et al., 2013; Taylor-Harding et al., 2015). Such cells, although they may, or may not, show sensitivity to MYC inhibition, would not be the best models to test synergies of MYC addition and inhibitor sensitivity with CDK4/6 inhibition, because of their intrinsic resistance to the latter intervention.

On the other hand, a search in public databases (www.cbioportal.org) shows that cell lines (877 cell lines in the Novartis-Broad Institute Cancer Cell line Encyclopedia) with homozygous RB1 loss tend not to have altered gene copy numbers or expression levels of MYC. Conversely, amplification and/or overexpression of MYC tend not to associate with RB1 loss. In these datasets, amplification of cyclin E is most often not associated with amplification and/or overexpression of MYC, although the mutual exclusivity is not as significant as the MYC-RB1 mutual exclusivity. This suggests that these two genetic alterations (RB1 loss and cyclin E amplification) are not linked to MYC mRNA overexpression and thus possibly not associated with MYC addiction.

Responses to Reviewer #3:

The manuscript by Tarrado-Castellarnau et al. reports a metabolic reprogramming induced by Cdk4/6 inhibition. In particular, CDK4/6 knockdown results in increased glycolysis accompanied by enhanced glutamine oxidation and contribution to the oxidative TCA cycle, as well as increased concentration of TCA intermediates. Glutamine seems the preferred substrate for respiration. Interestingly, these changes are associated to MYC accumulation and a MYC-dependent transcriptional response leading to increased glutamine metabolism, mTOR activation and HIF1 α -mediated responses. As a result, these cells become sensitive to inhibitors of these pathways, suggesting new combinatory approaches to improve the use of CDK4/6 inhibitors in the clinic. In general, the manuscript describes a number of very interesting observations regarding the metabolic changes induced by CDK4/6 inhibition using either RNAi or palbociclib, a CDK4/6 specific inhibitor recently approved for treating hormone-positive breast cancer. The quality of the metabolic analysis is very high and, in general, this work provides the most complete analysis of the metabolic changes induced by CDK4/6 inhibition so far.

On the other hand, the quality of the cell biology analysis is not as high as the metabolic part and the authors should improve some of the aspects of the manuscript as describe below.

Major points

1. Role of MYC in the phenotypes observed. The authors attribute to MYC accumulation most of the metabolic changes: see for instance "by identifying the accumulation of MYC as the major upstream event that explains most aspects of the metabolic reprogramming that follows CDK4/6 inhibition." (pag. 16) or "depletion or inhibition of CDK4/6 in cancer cells leads to de novo addiction to MYC" (page 17). However, the role of MYC as a major CDK4/6 target is not demonstrated. The early evidences that MYC peptides are CDK4 substrates (Anders et al., 2011) have very limited value and the relevance of a possible phosphorylation of MYC by CDK4/6 has not been explored in the past. Similarly, the exact phosphorylation site has not been analyzed and it is not clear why the authors use pSer62 as a read-out of CDK4/6 activity. In general, our knowledge on the relevance of MYC phosphorylation by CDK4/6 is too limited to draw any conclusion. There are many "indirect" reasons why MYC may be less phosphorylated apart from direct phosphorylation by CDK4/6.

In addition, the effect of MG132 is much more obvious than the effect of CDK4/6 inhibition and MG132 would lead to a similar effect in any protein degraded in a proteasome-dependent manner without indicating a direct effect. Thus, a more detailed analysis of the MYC-dependent phosphorylation by CDK4/6 is required for supporting the major conclusion in the manuscript. A direct manner to demonstrate the role of MYC would be to identifying MYC phosphorylation site and expressing phosphomimetic/phosphoresistant mutants to rescue/mimic the phenotypes induced by CDK4/6 inhibition.

We agree with the reviewer that this is a relevant issue. In response, we have performed two different kinase assays with CDK4-Cyclin D1 or CDK6-Cyclin D1 complexes and full-length recombinant human MYC protein as a substrate. RB protein was used as a positive kinase substrate control. MYC was clearly phosphorylated in both kinase reactions, indicating that CDK4-Cyclin D1 and CDK6-Cyclin D1 complexes directly phosphorylate MYC protein (Figure 5D).

The precise phosphorylation sites on MYC by CDK4/6-Cyclin D1 complexes were determined by performing kinase assays with unlabeled ATP followed by mass spectrometry analysis of MYC trypsin peptide fragments. This analysis yielded two specific phosphorylation sites on MYC, Thr58 and Ser62 (Figure EV3A).

We next determined P-Thr58 and P-Ser62 MYC levels in control and CDK4/6-inhibited cell lysates by means of Western blotting assays with antibodies specific for these two phosphosites. The results show clearly diminished levels of the two phosphorylated forms of MYC relative to total MYC upon inhibition of CDK4/6, which further supports that Thr58 and Ser62 are indeed substrates of CDK4/6 kinase activity also in live cells (Figure 5C).

Finally, as suggested by the referee, we expressed in HCT116 cells a T58A MYC mutant (Brady et al., 2014) (a gift from Christopher Counter; Addgene plasmid #53178), non-phosphorylatable on Thr58, thus mimicking constitutive CDK4/6 inhibition. We analyzed the resulting consumptions and productions rates of glucose, glutamine, lactate and glutamate and we observed the same increases in these rates as compared to control cells inhibited for CDK4/6 (Figure EV3B).

Together, these results confirm that CDK4/6-Cyclin D1 complexes phosphorylate MYC on Thr58 and Ser62, the two phosphorylation events required for MYC degradation through the proteasome (Gregory and Hann, 2000; Sears, 2000; Welcker et al., 2004).

We have detailed these new experiments in the Appendix Supplementary Methods (pages 9-10), described them in the Results (page 10) and discussed them in Discussion (page 17):

In order to test this hypothesis, we performed in vitro kinase assays with CDK4-Cyclin D1 or CDK6-Cyclin D1 complexes and full-length recombinant human c-MYC protein (Abcam, ab169901) as a substrate. Indeed, we detected ³³P signals in both kinase reactions, indicating that both CDK4-Cyclin D1 and CDK6-Cyclin D1 complexes directly phosphorylate MYC (Figure 5D). With the purpose of determining the precise phosphorylation sites, we performed kinase assays with unlabeled ATP and analyzed MYC tryptic peptides by mass spectrometry. The results showed that peptides KFELLPT(phosphor)PPLSPSR and KFELLPTPPLS(phosphor)PSRR were phosphorylated on Threonine 7 (corresponding to c-MYC T58) and Serine 11 (corresponding to c-MYC S62), respectively (Figure EV3A). Moreover, CDK4/6-kd cells displayed diminished P-MYC (Thr58)/MYC and P-MYC (Ser62)/MYC ratios compared to control cells (Figure 5C), supporting that phosphorylation of MYC at Thr58 and Ser62 is mediated by CDK4/6 in live cells. Consistently, cells expressing the MYC T58A phosphoresistant mutant mimicked the metabolic phenotype induced by CDK4/6 inhibition, as shown by enhancing glucose and glutamine consumption as well as lactate and glutamate production (Figure EV3B). Collectively, these observations suggest that CDK4/6-dependent phosphorylation is associated with the polyubiquitination and subsequent proteasomal

degradation of MYC, thus offering a plausible mechanism for the accumulation of MYC upon inhibition of CDK4/6.

Discussion (page 17):

We provide evidence that MYC is a phosphorylation substrate of CDK4/6-Cyclin D1 complexes at Thr58 and Ser62, and depletion of CDK4/6 in HCT116 cells prevents the phosphorylation of MYC at these two residues and its proteasome-mediated degradation (Sears, 2000).

2. Both glycolysis (ECAR) and oxidative respiration (OCR) seem to be increased upon CDK4/6 knockdown. It seems that CDK4/6 inhibition may trigger a general energetic-stress response rather than specifically modulate glycolysis. Thus, one of the most important concerns in the manuscript is to what extent the observations are caused by direct CDK4/6 knock-down/inhibition or by cell cycle arrest. To address this question the authors use thymidine-arrested cells. This is not the right control. Thymidine treatment results in defective S-phase and in an early S-phase arrest, whereas inhibition of CDK4/6 should arrest cells in G0-early/mid G1. This is a crucial point as all the phenotypes observed may be a consequence of G0/G1 arrest. The authors should compare CDK4/6 inhibition with a pure G0 arrest, for instance, serum deprivation versus CDK4/6 inhibition in non-transformed cells.

We thank the reviewer for raising this issue. In response, we performed experiments with serum deprivation on HCT116 human colorectal carcinoma cells and non-transformed NCM460 human epithelial cells derived from healthy colon mucosa (Moyer et al., 1996), as suggested by the reviewer. We found that serum deprivation induced a greater G0/G1 arrest than CDK4/6 inhibition, as seen in Figure 1D and Figure EV1G compared to Figure 5K for HCT116 cells and in Figure EV4G for NCM460 cells. Our results showed that in both HCT116 and NCM cell lines, cells synchronized in the G1 phase of the cell cycle presented comparable protein levels of MYC, GLS1, P-mTOR and HIF-1 α to asynchronous cells. We also measured the consumption and production rates of glucose, glutamine, lactate and glutamate in serum-deprived and control cells. HCT116 G0/G1 synchronized cells presented a statistically significant decrease of all these rates while NCM460 serum-starved cells did not change their consumption and production rates compared to control cells. Collectively, these results support the conclusion that the modulation of MYC, GLS1, P-mTOR and HIF-1 α that follows CDK4/6 depletion or inhibition is not a result of G1 arrest but directly attributable to a loss of function of CDK4/6. We have included these findings in the results section (page 13, paragraph 3):

Cells synchronized in the G1 phase of the cell cycle (Figure 5K) did not accumulate MYC, GLS1 or P-mTOR as compared to asynchronous cells, and presented higher HIF-1 α levels than cells with CDK4/6 inhibition (Figure 5L). We found the same results with NCM460 human epithelial cells derived from healthy colon mucosa (Figure EV4G-H). Further, the consumption and production rates of glucose, glutamine, lactate and glutamate in G1-synchronized cells were significantly lower than those of control HCT116 cells (Figure 5M) and not significantly different from asynchronous NCM460 cells (figure EV4I). These observations support the conclusion that the modulation of MYC, GLS1, P-mTOR and HIF-1 α that follows CDK4/6 depletion or inhibition is not a result of G1 arrest but directly attributable to a loss of function of CDK4/6.

3. Similarly, the characterization of the TOR response to CDK4/6 inhibition is limited to discussion of published data. CDK4/6 are known to directly phosphorylate TSC2 and this may explain some phenotypes presented in the manuscript.

The reported phosphorylation of TSC1 and TSC2 as substrates of CDK4/6 (Zacharek et al., 2005) leads to the inhibition of its activity as an inhibitor of RHEB activation, thus leading to mTOR activation (Inoki et al., 2003). Zacharek et al. further demonstrated that CDK4/6 inhibition leads to active TSC2 with consequent mTOR inhibition. This outcome is the exact opposite of our findings (activation of mTOR upon CDK4/6 inhibition), and thus a mechanism independent of a direct CDK4/6 regulation of mTOR must be invoked in our system.

Our study provides the following evidences in support of a MYC-driven activation of mTOR upon CDK4/6 inhibition in the cancer cell lines under study:

(1) MYC is known to downregulate the mTOR negative regulator TSC2 (Ravitz et al., 2007), as also found in our experiments with CDK4/6 inhibition (Figure 5G).

(2) The enhanced glutaminolysis that follows MYC upregulation would also contribute to the observed enhanced mTOR activation (Duran et al., 2012).

(3) In our experiments, inhibition or knockdown of MYC results in the reversal of the baseline activation of mTOR (phospho-mTOR, phospho-S6K) induced by inhibition of CDK4/6 in HCT116 cells (Figure 6B,E). This places MYC upstream of mTOR in this system. In consequence, we believe that our proposal that the upregulation of MYC following CDK4/6 inhibition is a sound mechanism to explain the observed mTOR upregulation. We have included in the Discussion section an explanation addressing this issue (page 18, paragraph 2):

We have also found that CDK4/6 inhibition or knockdown is accompanied with a baseline activation of the mTOR signaling pathway in line with prior evidences (Franco et al., 2016). It has been described that the mTOR negative regulators TSC1 and TSC2 are phosphorylated and inactivated by CDK4/6, and inhibition of CDK4/6 leads to active TSC1 and TSC2 and consequent inhibition of mTOR (Zachareck et al., 2005). However, inhibition of CDK4/6 in our experiments leads to enhanced, not diminished, baseline activation of mTOR and thus a mechanism independent of a direct CDK4/6 regulation of mTOR must be invoked in order to explain our observations. Our study provides the following evidences in support of a MYC-driven activation of mTOR: First, MYC is known to downregulate the mTOR negative regulator TSC2 (Ravitz et al., 2007), as also found in our experiments. Second, the enhanced glutaminolysis that follows MYC upregulation, in turn mediated by downstream upregulation of SCL1A5 and GLS (Gao et al., 2009), would also contribute to the observed enhanced mTOR activation (Duran et al., Mol Cell 47:349-358, 2012)., In consequence, we believe that our proposal that the upregulation of MYC following CDK4/6 inhibition is a sound mechanism to explain the observed mTOR upregulation.

4. Same rationale applies to P-Akt, a typical mark downstream of several feedback loops downstream of TOR. In general, it is not clear to what extent the Akt-TSC2-TOR is modified as a direct target or a secondary consequence of the alterations induced by CDK4/6 knockdown/inhibition.

The upregulation of mTOR activity upon CDK4/6 inhibition, observed in our results, is expected to lead to activation of TORC2, which would explain the enhanced phosphorylation of Akt on Ser473 (Sarbasov et al., 2005) observed in our experiments. Phosphorylation at the C-terminal domain of Akt (including Ser473) provides enhanced stability to the protein and thus permits full activation of the enzymatic activity of Akt upon phosphorylation of its activation loop site (including Thr308). Phosphorylation of Thr308 and other activation loop residues is mediated by PDK, in turn activated by PI3K and upstream signaling (reviewed in (Risso et al., 2015)). Therefore, we suggest that the activation of Akt-TSC2-mTOR is an indirect/secondary consequence of CDK4/6 inhibition.

5. Similarly, It is very difficult in the manuscript to conclude on to what extent the upregulation of MYC is responsible for the changes in TOR and metabolic pathways in the cell. Or similarly, whether the changes in TOR are responsible for the metabolic changes.

We agree with the reviewer that downstream metabolic consequences of high MYC and mTOR levels show a significant overlap and thus it may be difficult to discriminate whether MYC or mTOR are the relevant upstream events. We would like to argue that, of these two alternatives, the likely upstream event in the system under study is MYC, based on the following:

- (1) In our experiments, inhibition or knockdown of MYC results in the reversal of the baseline activation of mTOR (phospho-mTOR, phospho-S6K) induced by inhibition of CDK4/6 in HCT116 cells (Figure 6B, 6E). This places MYC upstream of mTOR in this system.
- (2) Despite overlaps between MYC target genes and mTOR regulated genes, metabolically relevant genes including glutaminase (GLS) or the glutamine transporter SLC7A5, are characteristic MYC target genes (Gao et al., 2009) but not known to be directly regulated by mTOR. The transcripts for these genes are significantly upregulated upon CDK4/6 inhibition (Figure 4C and 5F) and their encoded proteins and activities are strong contributors to the overall adaptive metabolic phenotype.

Other points

6. Lack of p16 does not automatically results in "constitutive activation of CDK4/6" (page 4). These kinases still require binding to the cyclin, activating phosphorylation, etc.

We thank the reviewer for this comment. We have modified the phrasing of the corresponding sentence as follows (page 4, 1st paragraph of the Results section):

These cells bear a loss-of-function p16^{INK4a} mutant allele and a wild type allele silenced through an hypermethylated promoter, resulting in full loss of functional p16^{INK4a} (Myohanen et al., 1998), which can lead to a higher activation status of CDK4/6.

7. CDK4/6 knockdown results in increased apoptosis (Fig. 1C). This is somehow unexpected for CDK4/6 knockdown. Any hypothesis for this? Are the authors sure about the specificity of the RNAi reagents?

In our experiments, we have observed relatively modest but consistent levels of apoptosis (about 5%) after RNAi-mediated depletion of CDK4/6 (Figure 1C), which suggests that the observed effect is unlikely to be an off-target consequence of RNAi (in comparison, transfection of non-targeting RNA duplexes cause 1-2% apoptosis). On the other hand it has been previously described by others that knockdown of CDK4 (Hagen et al., 2013; Retzer-Lidl et al., 2007) and CDK6 (Zhang et al., 2014) using RNAi induces a modest level of apoptosis. Likewise, inhibition of CDK4/6 with palbociclib has also been reported to cause apoptosis of T cell acute lymphoma cells (Choi et al., 2012; Sawai et al.). Although to our knowledge the mechanism of this low-level cell death upon CDK4/6 inhibition has not been explored, we speculate that it may be a consequence of the enhanced levels of ROS undergone by CDK4/6-inhibited cells (Figure EV2B).

8. Not sure if the effects in Figure 7 represent synergy or cooperation. This should be tested and discussed appropriately.

To elucidate whether the effects of drug combinations represent synergy or cooperation as well as to quantitatively determine the synergy of the dose-dependent effect on cell viability, the Combination Index (CI) equation of Chou and Talalay (Chou and Talalay, 1984) was used with the CompuSyn software (ComboSyn, Inc., Paramus, NJ, USA). The CI equation determines the additive effect of drug combinations, such that synergism is defined as a greater-than-expected-additive effect, and antagonism is defined as less-than-expected-additive effect. Thus, CI = 1 indicates an additive (cooperative) effect, CI < 1 indicates a synergistic effect, and CI > 1 indicates antagonism. CI values are interpreted as follows (Reynolds and Maurer, 2005):

CI value	Agonistic effect
<0.10	Very strong synergism
0.10–0.30	Strong synergism
0.30–0.70	Synergism
0.70–0.90	Moderate to slight synergism
0.90–1.10	Nearly additive
1.10–1.45	Slight to moderate antagonism
1.45–3.30	Antagonism
>3.30	Strong to very strong antagonism

The CI results obtained with CompuSyn for each combination of drug doses tested are listed in Table EV4. Synergistic antiproliferative effects of combined treatments. In all cases, synergy (CI < 1) was found in the antiproliferative effects of the combined treatments at each dose combination tested.

9. Why CDK4/6 knockdown reduces the anchorage-independent growth of HCT116 cells? A role of CDK4/6 in anchorage -independent growth as not been proposed in the past.

We would like to point out that a number of prior evidences support the role played by CDK4/6 – Cyclin D1-3 in conferring anchorage-independent growth to tumor (Arber et al., 1997; Gan et al., 2009), undergoing reprogramming into pluripotent cells (Tanabe et al., 2013) and non-tumor (Chen et al., 2003) cells.

10. Fig. 1M uses palbociclib whereas the rest of the data have been generated with CDK4/6 knock-down.

We have performed all the experiments using siRNA techniques except the cell proliferation assays, in which we tested drug combinations at a wide dose range. However, in the case of internal metabolite relative quantification by GC/MS, we need to work with a large number of cells ($>10^7$ cells) in order to obtain sharp and intense peaks for intermediates like citrate or α -ketoglutarate. To obtain this number of cells using siRNA techniques is not feasible for us, both experimentally and financially. For this reason, in this approach we used palbociclib which enables to work with higher number of cells, guaranteeing the detection of low concentration intermediates.

11. Figure 1B (S1A). Does "proliferation" means number of cells? The definition of these variables is not clear in the figures. For instance in Fig. S1C the authors indicate "survival" in the y axis but describe proliferation in the figure legend.

We thank the reviewer for bringing this to our attention. We have changed the designation of the y axis of Figures 1B and EV1F to "Cell number (% of control)", Figure EV1E to "Cell viability (%)" and Figure 6A to "Cell number (final-initial) (% of 0 μ M 10058-F4)".

12. Figure 1D (S1B). FACS (PI) profiles do not indicate percentage of G1 or G2/M cells but DNA content (2n, 4n, etc)

Figure 1D, Figure EV1G, Figure EV4G show the estimated percentage of cells at each cell cycle phase after cell cycle analysis of the flow cytometric DNA content histograms using the Multicycle software (Phoenix Flow Systems, San Diego, CA, USA) which applies the algorithm described by (Rabinovitch, 1994). The DNA content histogram related to Figure 1D and Figure EV1G is shown in Figure EV1C and EV1J, respectively.

13. Fig. S1A could be combined with Fig. 1B.

Due the strong relationship between Figure S1A and all the other panels of Figure S1, we believe it is to reader's benefit to group them in the same figure. We have incorporated all the panels in Figure S1 to the Expanded View Figure 1 in order to make these results more accessible to the reader.

14. The authors may want to discuss recent papers (see for instance papers by E. Knudsen or N. Dyson) on the effect of palbociclib (or Cdk4-pRb) in mitochondrial function.

We have included a reference on the effect of pRB loss in the discussion, as suggested by the referee (page 18, paragraph 1):

Consistent with our results, pRB inhibition, whose phenotype is expected to be the opposite to CDK4/6 loss, has been described to be associated with a decrease in TCA cycle and mitochondrial respiration (Nicolay et al., 2015).

We have linked the results presented by Franco et al. with ours (page 7, paragraph 2):
CDK4/6-kd cells exhibited higher oxygen consumption rates (OCR) than control cells (Figure 2A-B), indicating an augmented mitochondrial respiration which is in agreement with recently published studies (Franco et al., 2016).

And discussed their findings (page 17, paragraph 1):
A recent study has found that CDK4/6 inhibition leads to an enhanced activation of the mTOR pathway and consequent sensitivity of cancer cells to mTOR inhibitors (Franco et al., 2016).

Page 18, paragraph 2:

We have also found that CDK4/6 inhibition or knockdown is accompanied with a baseline activation of the mTOR signaling pathway in line with prior evidences (Franco et al., 2016).

Revision references:

- Arber, N., Doki, Y., Han, E. K., Sgambato, A., Zhou, P., Kim, N. H., Delohery, T., Klein, M. G., Holt, P. R., and Weinstein, I. B. (1997). Antisense to cyclin D1 inhibits the growth and tumorigenicity of human colon cancer cells. *Cancer Res* 57, 1569-1574.
- Brady, D. C., Crowe, M. S., Turski, M. L., Hobbs, G. A., Yao, X., Chaikuad, A., Knapp, S., Xiao, K., Campbell, S. L., Thiele, D. J., and Counter, C. M. (2014). Copper is required for oncogenic BRAF signalling and tumorigenesis. *Nature* 509, 492-496.
- Brunelle, J. K., Santore, M. T., Budinger, G. R., Tang, Y., Barrett, T. A., Zong, W. X., Kandel, E., Keith, B., Simon, M. C., Thompson, C. B., *et al.* (2004). c-Myc sensitization to oxygen deprivation-induced cell death is dependent on Bax/Bak, but is independent of p53 and hypoxia-inducible factor-1. *The Journal of biological chemistry* 279, 4305-4312.
- Chen, Q., Lin, J., Jinno, S., and Okayama, H. (2003). Overexpression of Cdk6-cyclin D3 highly sensitizes cells to physical and chemical transformation. *Oncogene* 22, 992-1001.
- Choi, Yoon J., Li, X., Hydbring, P., Sanda, T., Stefano, J., Christie, Amanda L., Signoretti, S., Look, A. T., Kung, Andrew L., von Boehmer, H., and Sicinski, P. (2012). The Requirement for Cyclin D Function in Tumor Maintenance. *Cancer cell* 22, 438-451.
- Chou, T. C., and Talalay, P. (1984). Quantitative analysis of dose-effect relationships: the combined effects of multiple drugs or enzyme inhibitors. *Adv Enzyme Regul* 22, 27-55.
- Duran, R. V., Oppliger, W., Robitaille, A. M., Heiserich, L., Skendaj, R., Gottlieb, E., and Hall, M. N. (2012). Glutaminolysis activates Rag-mTORC1 signaling. *Mol Cell* 47, 349-358.
- Edgar, R., Domrachev, M., and Lash, A. E. (2002). Gene Expression Omnibus: NCBI gene expression and hybridization array data repository. *Nucleic Acids Res* 30, 207-210.
- Franco, J., Balaji, U., Freinkman, E., Witkiewicz, Agnieszka K., and Knudsen, Erik S. (2016). Metabolic Reprogramming of Pancreatic Cancer Mediated by CDK4/6 Inhibition Elicits Unique Vulnerabilities. *Cell Reports*.
- Gan, L., Liu, P., Lu, H., Chen, S., Yang, J., McCarthy, J. B., Knudsen, K. E., and Huang, H. (2009). Cyclin D1 promotes anchorage-independent cell survival by inhibiting FOXO-mediated anoikis. *Cell Death and Differentiation* 16, 1408-1417.
- Gao, P., Tchernyshyov, I., Chang, T. C., Lee, Y. S., Kita, K., Ochi, T., Zeller, K. I., De Marzo, A. M., Van Eyk, J. E., Mendell, J. T., and Dang, C. V. (2009). c-Myc suppression of miR-23a/b enhances mitochondrial glutaminase expression and glutamine metabolism. *Nature* 458, 762-765.
- Gregory, M. A., and Hann, S. R. (2000). c-Myc proteolysis by the ubiquitin-proteasome pathway: stabilization of c-Myc in Burkitt's lymphoma cells. *Molecular and cellular biology* 20, 2423-2435.
- Hagen, K. R., Zeng, X., Lee, M. Y., Tucker Kahn, S., Harrison Pitner, M. K., Zaky, S. S., Liu, Y., O'Regan, R. M., Deng, X., and Saavedra, H. I. (2013). Silencing CDK4 radiosensitizes breast cancer cells by promoting apoptosis. *Cell Div* 8, 10.
- Herrera-Abreu, M. T., Palafox, M., Asghar, U., Rivas, M. A., Cutts, R. J., Garcia-Murillas, I., Pearson, A., Guzman, M., Rodriguez, O., Grueso, J., *et al.* (2016). Early Adaptation and Acquired Resistance to CDK4/6 Inhibition in Estrogen Receptor-Positive Breast Cancer. *Cancer Res* 76, 2301-2313.
- Huang, M. J., Cheng, Y. C., Liu, C. R., Lin, S., and Liu, H. E. (2006). A small-molecule c-Myc inhibitor, 10058-F4, induces cell-cycle arrest, apoptosis, and myeloid differentiation of human acute myeloid leukemia. *Exp Hematol* 34, 1480-1489.
- Inoki, K., Li, Y., Xu, T., and Guan, K. L. (2003). Rheb GTPase is a direct target of TSC2 GAP activity and regulates mTOR signaling. *Genes Dev* 17, 1829-1834.
- Jansen, V. M., Bhola, N. E., Bauer, J. A., Formisano, L., Lee, K. M., Hutchinson, K. E., Witkiewicz, A. K., Moore, P. D., Estrada, M. V., Sanchez, V., *et al.* (2017). Kinome-wide RNA interference screen reveals a role for PDK1 in acquired resistance to CDK4/6 inhibition in ER-positive breast cancer. *Cancer Res*.
- Konecny, G. E., Winterhoff, B., Kolarova, T., Qi, J., Manivong, K., Dering, J., Yang, G., Chalukya, M., Wang, H. J., Anderson, L., *et al.* (2011). Expression of p16 and retinoblastoma determines response to CDK4/6 inhibition in ovarian cancer. *Clin Cancer Res* 17, 1591-1602.
- Koshiji, M., Kageyama, Y., Pete, E. A., Horikawa, I., Barrett, J. C., and Huang, L. E. (2004). HIF-1alpha induces cell cycle arrest by functionally counteracting Myc. *Embo J* 23, 1949-1956. Epub 2004 Apr 1948.
- Koshiji, M., To, K. K., Hammer, S., Kumamoto, K., Harris, A. L., Modrich, P., and Huang, L. E. (2005). HIF-1alpha induces genetic instability by transcriptionally downregulating MutSalpalpha expression. *Mol Cell* 17, 793-803.
- Lin, C. P., Liu, J. D., Chow, J. M., Liu, C. R., and Liu, H. E. (2007). Small-molecule c-Myc inhibitor, 10058-F4, inhibits proliferation, downregulates human telomerase reverse transcriptase

- and enhances chemosensitivity in human hepatocellular carcinoma cells. *Anticancer Drugs* 18, 161-170.
- Logan, J. E., Mostofizadeh, N., Desai, A. J., E, V. O. N. E., Conklin, D., Konkankit, V., Hamidi, H., Eckardt, M., Anderson, L., Chen, H. W., *et al.* (2013). PD-0332991, a potent and selective inhibitor of cyclin-dependent kinase 4/6, demonstrates inhibition of proliferation in renal cell carcinoma at nanomolar concentrations and molecular markers predict for sensitivity. *Anticancer Res* 33, 2997-3004.
- Moyer, M. P., Manzano, L. A., Merriman, R. L., Stauffer, J. S., and Tanzer, L. R. (1996). NCM460, a normal human colon mucosal epithelial cell line. *In Vitro Cell Dev Biol Anim* 32, 315-317.
- Myohanen, S. K., Baylin, S. B., and Herman, J. G. (1998). Hypermethylation can selectively silence individual p16ink4A alleles in neoplasia. *Cancer Res* 58, 591-593.
- Nicolay, B. N., Danielian, P. S., Kottakis, F., Lapek, J. D., Sanidas, I., Miles, W. O., Dehnad, M., Tschöp, K., Gierut, J. J., Manning, A. L., *et al.* (2015). Proteomic analysis of pRb loss highlights a signature of decreased mitochondrial oxidative phosphorylation. *Genes & Development* 29, 1875-1889.
- Rabinovitch, P. S. (1994). DNA content histogram and cell-cycle analysis. *Methods Cell Biol* 41, 263-296.
- Ravitz, M. J., Chen, L., Lynch, M., and Schmidt, E. V. (2007). c-myc Repression of TSC2 contributes to control of translation initiation and Myc-induced transformation. *Cancer Research* 67, 11209-11217.
- Retzer-Lidl, M., Schmid, R. M., and Schneider, G. (2007). Inhibition of CDK4 impairs proliferation of pancreatic cancer cells and sensitizes towards TRAIL-induced apoptosis via downregulation of survivin. *International Journal of Cancer* 121, 66-75.
- Reynolds, C. P., and Maurer, B. J. (2005). Evaluating response to antineoplastic drug combinations in tissue culture models. *Methods Mol Med* 110, 173-183.
- Risso, G., Blaustein, M., Pozzi, B., Mammi, P., and Srebrow, A. (2015). Akt/PKB: one kinase, many modifications. *The Biochemical journal* 468, 203-214.
- Sarbasov, D. D., Guertin, D. A., Ali, S. M., and Sabatini, D. M. (2005). Phosphorylation and regulation of Akt/PKB by the rictor-mTOR complex. *Science (New York, NY)* 307, 1098-1101.
- Sawai, C. M., Freund, J., Oh, P., Ndiaye-Lobry, D., Bretz, Jamieson C., Strikoudis, A., Genesca, L., Trimarchi, T., Kelliher, Michelle A., Clark, M., *et al.* Therapeutic Targeting of the Cyclin D3:CDK4/6 Complex in T Cell Leukemia. *Cancer cell* 22, 452-465.
- Sears, R. (2000). Multiple Ras-dependent phosphorylation pathways regulate Myc protein stability. *Genes & Development* 14, 2501-2514.
- Tanabe, K., Nakamura, M., Narita, M., Takahashi, K., and Yamanaka, S. (2013). Maturation, not initiation, is the major roadblock during reprogramming toward pluripotency from human fibroblasts. *Proceedings of the National Academy of Sciences of the United States of America* 110, 12172-12179.
- Taylor-Harding, B., Aspuria, P. J., Agadjanian, H., Cheon, D. J., Mizuno, T., Greenberg, D., Allen, J. R., Spurka, L., Funari, V., Spiteri, E., *et al.* (2015). Cyclin E1 and RTK/RAS signaling drive CDK inhibitor resistance via activation of E2F and ETS. *Oncotarget* 6, 696-714.
- Tennant, D. A., Frezza, C., MacKenzie, E. D., Nguyen, Q. D., Zheng, L., Selak, M. A., Roberts, D. L., Dive, C., Watson, D. G., Aboagye, E. O., and Gottlieb, E. (2009). Reactivating HIF prolyl hydroxylases under hypoxia results in metabolic catastrophe and cell death. *Oncogene* 28, 4009-4021.
- Vora, S. R., Juric, D., Kim, N., Mino-Kenudson, M., Huynh, T., Costa, C., Lockerman, E. L., Pollack, S. F., Liu, M., Li, X., *et al.* (2014). CDK 4/6 inhibitors sensitize PIK3CA mutant breast cancer to PI3K inhibitors. *Cancer cell* 26, 136-149.
- Welcker, M., Orian, A., Jin, J., Grim, J. E., Harper, J. W., Eisenman, R. N., and Clurman, B. E. (2004). The Fbw7 tumor suppressor regulates glycogen synthase kinase 3 phosphorylation-dependent c-Myc protein degradation. *Proceedings of the National Academy of Sciences of the United States of America* 101, 9085-9090.
- Yin, X., Giap, C., Lazo, J. S., and Prochownik, E. V. (2003). Low molecular weight inhibitors of Myc-Max interaction and function. *Oncogene* 22, 6151-6159.
- Zacharek, S. J., Xiong, Y., and Shumway, S. D. (2005). Negative regulation of TSC1-TSC2 by mammalian D-type cyclins. *Cancer Res* 65, 11354-11360.
- Zhang, H., Gao, P., Fukuda, R., Kumar, G., Krishnamachary, B., Zeller, K. I., Dang, C. V., and Semenza, G. L. (2007). HIF-1 inhibits mitochondrial biogenesis and cellular respiration in VHL-deficient renal cell carcinoma by repression of C-MYC activity. *Cancer cell* 11, 407-420.

Zhang, Z., Huang, L., Yu, Z., Chen, X., Yang, D., Zhan, P., Dai, M., Huang, S., Han, Z., and Cao, K. (2014). Let-7a functions as a tumor suppressor in Ewing's sarcoma cell lines partly by targeting cyclin-dependent kinase 6. *DNA Cell Biol* 33, 136-147.

Zirath, H., Frenzel, A., Oliynyk, G., Segerstrom, L., Westermarck, U. K., Larsson, K., Munksgaard Persson, M., Hultenby, K., Lehtio, J., Einvik, C., *et al.* (2013). MYC inhibition induces metabolic changes leading to accumulation of lipid droplets in tumor cells. *Proceedings of the National Academy of Sciences of the United States of America* 110, 10258-10263.

2nd Editorial Decision

06 July 2017

Thank you again for submitting your work to Molecular Systems Biology. We have now finally heard back from the two referees who accepted to evaluate the revised study. As you will see, referee #1 is now fully supportive. Reviewer #2 is however less positive. We note that reviewer #2 feels that the previous study by Franco et al should be presented more fairly and still questions the novelty of the study. Given that the other reviewers were more positive, also in the first round, and given that we are now after one round of major revision, we would not like to take novelty as a ground for rejection. It is however absolutely crucial to discuss this prior study in a transparent and fair way and we would invite you to amend the text in this direction. Reviewer #2 also feels that the importance of the metabolic analysis is exaggerated and we would thus ask you to tone down this aspect appropriately.

We would also ask you to add a formal Data and Software Availability Section after Materials & Methods.

 REVIEWER REPORTS

Reviewer #1:

I believe that the authors have substantively address most keep issues raised.

Reviewer #2:

My major concerns were only partly addressed. My concerns are reported again below. In contrast, all minor points were resolved.

1) Novelty. The rebuttal of the authors confirmed my opinion that the only novel aspect of this submission is that the effect of CDK4/6 inhibition is mediated by MYC. The authors did a substantial effort to expand the body of evidence, by assaying phosphorylation of MYC. Alas, this doesn't fix the issue that Franco et al already reported (i) increase in metabolic consumption of glucose, glutamine, and oxygen upon CDK4/6 inhibition in p16INK4a^{-/-} cells, (ii) increase in mitochondrial activity (and mass), and (iii) synthetic lethality with mTOR.

Considering that the paper by the Knudsen lab has been online since 2015, it is unacceptable that the authors purposely neglect the preexistence of a large study on this topic and invest 10 pages to basically replicate the same study. Specifically, the Franco paper has only been mentioned in the introduction in the general context of "tumor metabolic reprogramming is emerging as a key component of adaptive drug-induced stress that may unveil novel actionable vulnerabilities of cancer cells", whereas all aspects related to metabolic reprogramming of CDK4/6 inhibition, mitochondria, and mTOR vulnerability were omitted. To put it mildly, it is unfair.

2) One additional novel aspect of this submission is the detailed metabolic characterization of CDK4/6-KD cells presented in the first half of the text. I stand to the opinion that the relevance of the metabolic analysis part is overblown. The relevance of network biology is marginal. Novel insights are derived from traditional reasoning and experiments.

On a side note, there is a bit of confusion on what metabolite levels of fluxes can reveal. The argument "The relevance of changes in fluxes can be assessed by observing changes in metabolic intermediates, which are a function of the relative changes in metabolite supply and demand capacities. A change in the relative supply and demand maximal rates for a given metabolite that causes large fluctuations in the concentrations of such a metabolite may be accompanied with relatively small changes in the corresponding reaction fluxes. For instance, the measured increase in OCR by a factor of 1.3 that follows CDK4/6 inhibition would represent a significant increase in the production of ATP and ROS." is partly wrong and irrelevant. There is no simple link between levels and fluxes, for sure not for such highly connected metabolites

The magnitude of all TCA flux changes are minor, and fail to explain what causes the dependency on glutamine. In the rebuttal, the authors mention their best example "The estimated decrease in the flux through malate dehydrogenase is very significant if compared with the increases in glycolysis and mitochondrial respiration." Well: the different in MDH flux is $0.07 - 0.04 = 0.03$, glycolysis is $>20x$ higher (!). Apart of OXPHOS, there are many more reasons why glutamine could be important.

3) I wasn't able to verify whether the information has been made available because the supplementary data deposited on zenodo is not publicly accessible. If the authors will remove the lock and did what they wrote in the rebuttal, this issue is completely solved.

2nd Revision - authors' response

03 August 2017

Responses to the Editor comments:

We have included the following information in the Data and Software Availability section at the end of Materials & Methods:

The primary datasets and computer code produced in this study are available in the following databases:

Microarray data: Gene Expression Omnibus GSE84597

(<https://www.ncbi.nlm.nih.gov/geo/query/acc.cgi?acc=GSE84597>).

Modeling computer scripts: Zenodo database (<https://doi.org/10.5281/zenodo.546717>).

Responses to Reviewer #1:

I believe that the authors have substantively addressed most keep issues raised.

We thank the reviewer for the positive comments of our manuscript.

Responses to Reviewer #2:

My major concerns were only partly addressed. My concerns are reported again below. In contrast, all minor points were resolved.

1) Novelty. The rebuttal of the authors confirmed my opinion that the only novel aspect of this submission is that the effect of CDK4/6 inhibition is mediated by MYC. The authors did a substantial effort to expand the body of evidence, by assaying phosphorylation of MYC. Alas, this doesn't fix the issue that Franco et al already reported (i) increase in metabolic consumption of glucose, glutamine, and oxygen upon CDK4/6 inhibition in p16INK4a^{-/-} cells, (ii) increase in mitochondrial activity (and mass), and (iii) synthetic lethality with mTOR. Considering that the paper by the Knudsen lab has been online since 2015, it is unacceptable that the authors purposely neglect the preexistence of a large study on this topic and invest 10 pages to basically replicate the same study. Specifically, the Franco paper has only been mentioned in the introduction in the general context of "tumor metabolic reprogramming is emerging as a key component of adaptive drug-induced stress that may unveil novel actionable vulnerabilities of cancer cells", whereas all aspects related to metabolic reprogramming of CDK4/6 inhibition, mitochondria, and mTOR vulnerability were omitted. To put it mildly, it is unfair.

Even though we had already cited the 2016 paper by Franco *et al.* in three additional occasions in the results and discussion sections of our first revised manuscript, highlighting common

observations between their work and ours, and in agreement with the reviewer, our newly revised manuscript places even more emphasis on that paper, in the hope that this better represents prior work. It is not our intention, nor can it be, to minimize significant work by fellow scientists. Quite the contrary, we strongly support that any solid prior evidence should be given proper credit as a basis for moving beyond in any field of knowledge. We believe that our finding of the role played by upregulation of MYC as the key adaptive event in response to CDK4/6 depletion or inhibition is conceptually unprecedented and far-reaching. Although this is a significant novelty and contribution to the field that builds upon both the prior knowledge contributed by Franco *et al.*, and our own sets of data on metabolic analysis that show significant convergence with Franco *et al.*'s findings, we do wish to unambiguously emphasize the relevance of Franco *et al.*'s contributions. We hope that the present version in which we have carefully revised that Franco *et al.* results are properly highlighted and extensively discussed will be satisfactory. In the following, we detail the paragraphs where we have rewritten and/or included citations to appropriately discuss the prior studies in a transparent and fair manner:

We have rewritten the citation of Franco *et al.* work in the Introduction section (Page 3, paragraph 2 and page 4, paragraph 1):

To extend the benefits of CDK4/6 inhibition and to mitigate acquired resistance in cancer management, the adaptations of cancer cells to CDK4/6 inhibition need to be investigated by exploring not only altered signaling pathways but also their crosstalk with metabolic reprogramming. In this regard, tumor metabolic reprogramming is emerging as a key component of adaptive responses to drug-induced stress that may unveil novel actionable vulnerabilities of cancer cells (Maiso et al., 2015; Tarrado-Castellarnau et al., 2016). Specifically, recent work by Franco et al. has unveiled metabolic reprogramming events and actionable metabolic targets, in particular mTOR, in pancreatic cancer cells in response to palbociclib (Franco et al., 2016). Herein, we have undertaken a systematic study of the consequences on central carbon metabolism of the depletion or inhibition of CDK4/6 in cancer cells. By complementing metabolic analysis with transcriptomic data, we accurately depict relevant metabolic shifts associated with CDK4/6 depletion, revealing that the upregulation of MYC and its downstream network, which includes glutaminolysis and mTOR signaling, is both a direct consequence of, and a key adaptation to CDK4/6 inhibition.

In the Results section:

Page 7, paragraph 2 (included in the first revised manuscript):
CDK4/6-kd cells exhibited higher oxygen consumption rates (OCR) than control cells (Figure 2A-B), indicating an augmented mitochondrial respiration which is in agreement with recently published studies (Franco et al., 2016).

Page 7, paragraph 3 (included in this revision):
Collectively, these observations indicate that CDK4/6 knockdown increases mitochondrial metabolism through elevated utilization of glutamine and enhanced mitochondrial respiratory capacity, and are in agreement with Franco et al. results for a pancreatic cancer cell model (Franco et al., 2016). As such, specific metabolic reprogramming events in response to CDK4/6 depletion or inhibition appear to be conserved among cancer cells of different origin.

Page 9, paragraph 1 (included in this revision):
Gene expression profiling identified 1308 genes differentially expressed in CDK4/6-kd vs. control cells (718 upregulated, 592 downregulated; Figure 4A; see also Table EV2). Gene Set Enrichment Analysis (GSEA) (Subramanian et al., 2005) yielded a significant enrichment in CDK4/6-kd cells of signatures associated with MYC (upregulation), mechanistic target of rapamycin (mTOR) (upregulation) or hypoxia inducible factor 1 α (HIF-1 α) (downregulation) (Figure 4B; see also Table EV3). The inferred upregulation of the mTOR pathway is in agreement with previous studies (Franco et al., 2016).

Page 11, paragraph 2 (included in this revision):

Our analysis additionally suggested that CDK4/6 knockdown induced an upregulation of the mTOR pathway (Figure 4; see also Table EV3), an inference experimentally supported by the observation of increased levels of P-mTOR (Ser2448) in CDK4/6-kd cells under standard growth conditions (Figure 5E), in concordance with recent studies (Franco et al., 2016).

In the Discussion section:

Page 17, paragraph 2 (rewritten, included in the first revised manuscript):

Our metabolic and transcriptomic analyses have allowed us to precisely map the extensive metabolic reprogramming that follows CDK4/6 depletion in HCT116 colorectal cancer cells, characterized by an increase in mitochondrial metabolism and function accompanied with an enhanced metabolism of glucose, glutamine and amino acids. These observations are in agreement with those reported by Franco et al. (Franco et al., 2016), who also found that CDK4/6 inhibition leads to an enhanced activation of the mTOR pathway and consequent sensitivity of pancreatic cancer cells to mTOR inhibitors. Significantly, our analysis identifies the accumulation of MYC as the major upstream event that explains most aspects of the metabolic reprogramming that follows CDK4/6 inhibition, including upregulation of mTOR. We provide evidence that MYC is a direct phosphorylation substrate of CDK4/6-Cyclin D1 complexes at Thr58 and Ser62, and depletion of CDK4/6 in HCT116 cells prevents the phosphorylation of MYC at these two residues and its proteasome-mediated degradation (Sears, 2000). As discussed below in more detail, the consequent accumulation of MYC mechanistically explains the simultaneous greater glutaminase dependence, downstream upregulation of the mTOR pathway and blunting of cellular responses to hypoxia of treated cells.

Page 18, paragraph 2 (included in the first revised manuscript):

We have also found that CDK4/6 inhibition or knockdown is accompanied with a baseline activation of the mTOR signaling pathway, in line with prior evidences (Franco et al., 2016). It has been described that the mTOR negative regulators TSC1 and TSC2 are phosphorylated and inactivated by CDK4/6, and inhibition of CDK4/6 leads to active TSC1 and TSC2 and consequent inhibition of mTOR (Zacharek et al., 2005). However, inhibition of CDK4/6 in our experiments leads to enhanced, not diminished, baseline activation of mTOR and thus a mechanism independent of a direct CDK4/6 regulation of mTOR must be invoked in order to explain our observations.

2) One additional novel aspect of this submission is the detailed metabolic characterization of CDK4/6-KD cells presented in the first half of the text. I stand to the opinion that the relevance of the metabolic analysis part is overblown. The relevance of network biology is marginal. Novel insights are derived from traditional reasoning and experiments. On a side note, there is a bit of confusion on what metabolite levels of fluxes can reveal. The argument "The relevance of changes in fluxes can be assessed by observing changes in metabolic intermediates, which are a function of the relative changes in metabolite supply and demand capacities. A change in the relative supply and demand maximal rates for a given metabolite that causes large fluctuations in the concentrations of such metabolite may be accompanied with relatively small changes in the corresponding reaction fluxes. For instance, the measured increase in OCR by a factor of 1.3 that follows CDK4/6 inhibition would represent a significant increase in the production of ATP and ROS." is partly wrong and irrelevant. There is no simple link between levels and fluxes, for sure not for such highly connected metabolites.

The magnitude of all TCA flux changes are minor, and fail to explain what causes the dependency on glutamine. In the rebuttal, the authors mention their best example "The estimated decrease in the flux through malate dehydrogenase is very significant if compared with the increases in glycolysis and mitochondrial respiration.". Well: the different in MDH flux is $0.07-0.04 = 0.03$, glycolysis is $>20x$ higher (!). Apart of OXPHOS, there are many more reasons why glutamine could be important.

We have followed the reviewer's suggestions and in the current revised manuscript we have reduced the level of detail of our metabolic network analysis and appropriately toned down its importance (Page 8 paragraph 2):

To infer differential intracellular metabolic flux distributions in CDK4/6-kd vs. control cells, we constructed a quantitative metabolic network model of central carbon metabolism and applied a ¹³C metabolic flux analysis (Niedenfuhr et al., 2015) by applying the MATLAB-based software package INCA (Young, 2014). For this, we combined direct extracellular measurements, such as oxygen consumption, metabolite consumption and production rates and protein synthesis rate, with isotopologue distributions in several metabolites resulting from ¹³C propagation from labeled glucose and glutamine. The outcome of this analysis was compatible with all prior observations, supporting that CDK4/6-kd cells exhibit a higher mitochondrial activity accompanied with a higher dependence on glutamine and lower dependence on glucose (Figure 3 and Table EV1). In addition, the increased glutamine uptake and transformation to glutamate by glutaminase (GLS1) can also promote the mitochondrial respiratory activity since glutamate is the substrate for mitochondrial glutamate dehydrogenase (GDH). On the other hand, the extra demand of amino acids for protein synthesis matched the measured uptakes of essential amino acids in control cells, while CDK4/6-inhibited cells exhibited an extra uptake of several amino acids above the required value for protein synthesis (Appendix Figure S2).

In addition, as suggested by the reviewer, we have improved the Discussion section of the manuscript by highlighting the agreement of the metabolic reprogramming observed in colon adenocarcinoma cells with the prior observations by Franco *et al.*, who observed an increase in mitochondrial activity and metabolic consumptions of glucose, glutamine, and oxygen upon CDK4/6 inhibition in p16INK4a^{-/-} pancreas cancer cells (Page 17 paragraph 2 and page 18 paragraph 2):

Our metabolic and transcriptomic analyses have allowed us to precisely map the extensive metabolic reprogramming that follows CDK4/6 depletion in HCT116 colorectal cancer cells, characterized by an increase in mitochondrial metabolism and function accompanied with an enhanced metabolism of glucose, glutamine and amino acids. These observations are in agreement with those reported by Franco et al., who also found that CDK4/6 inhibition leads to an enhanced activation of the mTOR pathway and consequent sensitivity of pancreatic cancer cells to mTOR inhibitors (Franco et al., 2016). Significantly, our analysis identifies the accumulation of MYC as the major upstream event that explains most aspects of the metabolic reprogramming that follows CDK4/6 inhibition, including upregulation of mTOR. We provide evidence that MYC is a direct phosphorylation substrate of CDK4/6-Cyclin D1 complexes at Thr58 and Ser62, and depletion of CDK4/6 in HCT116 cells prevents the phosphorylation of MYC at these two residues and its proteasome-mediated degradation (Sears, 2000). As discussed below in more detail, the consequent accumulation of MYC mechanistically explains the simultaneous greater glutaminase dependence, downstream upregulation of the mTOR pathway and blunting of cellular responses to hypoxia of treated cells.

We also agree with the referee that in addition to oxidative phosphorylation there are other reasons why glutamine could be important. Accordingly, we have added a sentence mentioning this fact in the Results section of the manuscript:

Page 7, paragraph 3:

Collectively, these observations indicate that CDK4/6 knockdown increases mitochondrial metabolism through elevated utilization of glutamine and enhanced mitochondrial respiratory capacity, and are in agreement with Franco et al. results for a pancreatic cancer cell model (Franco et al., 2016). As such, specific metabolic reprogramming events in response to CDK4/6 depletion or inhibition appear to be conserved among cancer cells of different origin. Additional experiments showed that CDK4/6 depletion increased glutathione, NADPH and ROS levels, while it impaired fatty acid synthesis in HCT116 cells (Figure EV2), all of which are processes where glutamine is or can be involved.

We reason that the changes that we observe are relevant in that they are coordinated shifts in reactions that, combined, drive the TCA flux in a predominant direction. This fact can explain why our network analysis predicts a higher requirement of glutamine, in agreement with experimental data.

3) I wasn't able to verify whether the information has been made available because the supplementary data deposited on zenodo is not publicly accessible. If the authors will remove the lock and did what they wrote in the rebuttal, this issue is completely solved.

We apologize for the fact that the reviewer was unable to access Zenodo. We had included the reviewer access link for the files generated by INCA on the rebuttal letter but we had overlooked the fact that data were available through restricted access on the link displayed on the manuscript. In order to correct this problem, we have now made the data publicly accessible. As indicated in the revised manuscript, the data are publicly available at:
<https://doi.org/10.5281/zenodo.546717>

Revision references:

- Franco, J., Balaji, U., Freinkman, E., Witkiewicz, Agnieszka K., and Knudsen, Erik S. (2016). Metabolic Reprogramming of Pancreatic Cancer Mediated by CDK4/6 Inhibition Elicits Unique Vulnerabilities. *Cell Reports*.
- Maiso, P., Huynh, D., Moschetta, M., Sacco, A., Aljawai, Y., Mishima, Y., Asara, J. M., Roccaro, A. M., Kimmelman, A. C., and Ghobrial, I. M. (2015). Metabolic signature identifies novel targets for drug resistance in multiple myeloma. *Cancer Res* 75, 2071-2082.
- Niefenfuhr, S., Wiechert, W., and Noh, K. (2015). How to measure metabolic fluxes: a taxonomic guide for (13)C fluxomics. *Curr Opin Biotechnol* 34, 82-90.
- Sears, R. (2000). Multiple Ras-dependent phosphorylation pathways regulate Myc protein stability. *Genes & Development* 14, 2501-2514.
- Subramanian, A., Tamayo, P., Mootha, V. K., Mukherjee, S., Ebert, B. L., Gillette, M. A., Paulovich, A., Pomeroy, S. L., Golub, T. R., Lander, E. S., and Mesirov, J. P. (2005). Gene set enrichment analysis: a knowledge-based approach for interpreting genome-wide expression profiles. *Proceedings of the National Academy of Sciences of the United States of America* 102, 15545-15550.
- Tarrado-Castellarnau, M., Atauri, P. d., and Cascante, M. (2016). Oncogenic regulation of tumor metabolic reprogramming, Vol 7(38)).
- Young, J. D. (2014). INCA: a computational platform for isotopically non-stationary metabolic flux analysis. *Bioinformatics* 30, 1333-1335.
- Zacharek, S. J., Xiong, Y., and Shumway, S. D. (2005). Negative regulation of TSC1-TSC2 by mammalian D-type cyclins. *Cancer Res* 65, 11354-11360.

3rd Editorial Decision

15 August 2017

Thank you again for sending us your revised manuscript. We are now satisfied with the modifications made and I am pleased to inform you that your paper has been accepted for publication.

Corresponding Author Name: Marta Cascante

Manuscript Number: MSB-16-7321